# CONVERGENCE ANALYSIS OF HOMOTOPY-SGD FOR NON-CONVEX OPTIMIZATION

## ABSTRACT

First-order stochastic methods for solving large-scale non-convex optimization problems are widely used in many big-data applications, e.g. training deep neural networks as well as other complex and potentially non-convex machine learning models. Their inexpensive iterations generally come together with slow global convergence rate (mostly sublinear), leading to the necessity of carrying out a very high number of iterations before the iterates reach a neighborhood of a minimizer. In this work, we present a first-order stochastic algorithm based on a combination of homotopy methods and SGD, called Homotopy-Stochastic Gradient Descent (H-SGD), which finds interesting connections with some proposed heuristics in the literature, e.g. optimization by Gaussian continuation, training by diffusion, mollifying networks. Under some mild assumptions on the problem structure, we conduct a theoretical analysis of the proposed algorithm. Our analysis shows that, with a specifically designed scheme for the homotopy parameter, H-SGD enjoys a global linear rate of convergence to a neighborhood of a minimum while maintaining fast and inexpensive iterations. Experimental evaluations confirm the theoretical results and show that H-SGD can outperform standard SGD.

## 1  INTRODUCTION

This paper focuses on the theoretical development and analysis of a stochastic optimization algorithm, called Homotopy-Stochastic Gradient Descent (H-SGD), based on the combination of homotopy methods and stochastic gradient descent (SGD). The algorithm we propose is specifically designed to solve finite-sum problems of the following form

$$w^* \in \arg\min_{w \in \mathbb{R}^d} \left\{ f(w) := \frac{1}{N} \sum_{j=1}^{N} f_j(w) \right\}, \tag{1}$$

where $f : \mathbb{R}^d \to \mathbb{R}$ is continuously differentiable, bounded below and not necessarily convex. In particular, we assume that we only have access to noisy function values and gradients of the objective function in equation 1 via a stochastic first-order oracle, as in (Nemirovski et al., 2009) and (Ghadimi & Lan, 2013). Problems of this form typically arise in machine learning and deep learning applications, where the dimensionality of the datasets makes the full function and gradient evaluations too expensive. This class of problems is generally approximately solved by stochastic first-order iterative algorithms, e.g. SGD (Bottou et al., 2018), Adagrad (Duchi et al., 2011), Adam (Kingma & Ba, 2015). At the iteration $t$, the algorithms of this class acquire a stochastic estimate of the function value $f(w_t, \xi_t)$ and the gradient $g(w_t, \xi_t)$ by calling the oracle with input $w_t$, where $\xi_t$ is a random variable, i.e. when the noise comes from subsampling as in the mini-batch scenarios, then $\xi_t \in \{0,1\}^N$ with $\|\xi_t\|_1 = M$ and $g(w_t, \xi_t) = \frac{1}{M} \sum_{j=1}^{N} \xi_{t,j} \cdot \nabla f_j(w_t)$. In the case of SGD, for a given $w_0 \in \mathbb{R}^d$ and $\alpha > 0$, the iterates are generated as follows

$$w_{t+1} := w_t - \alpha g(w_t, \xi_t). \tag{2}$$

Consequently, the iterate $w_{t+1} = w_{t+1}(\xi_{[t]})$ is a function of the history $\xi_{[t]} := (\xi_0, \ldots, \xi_t)$ (also $w_0$ should be included in case it is a random initial point) of the generated random process and hence is itself random.

In general, stochastic first-order methods enjoy fast convergence when the problem is characterized by a certain structure. In particular, when the Polyak-Łojasiewicz (PL) condition (see Karimi et al., 2016, for more details on the PL condition) holds for the objective function in Problem 1, then, with a "small enough" value for the step-size, the SGD iterates converge linearly to a minimizer's neighborhood (Karimi et al., 2016; Vaswani et al., 2019). Unfortunately, in many machine learning applications, the PL condition is not a realistic assumption as the landscape is generally characterized by the presence of multiple local minima and saddle points (Dauphin et al., 2014; Kawaguchi, 2016; Karimi et al., 2017). At the same time, in the vicinity of the minimizers the problems generally show stronger structures, i.e. PL or even strong convexity, allowing for a faster local convergence (Karimi et al., 2017). In such a scenario, a smart initialization hence becomes crucial for the numerical performance of the method (Sutskever et al., 2013b). Unfortunately, the power of the existing smart initialization heuristics is often quite limited given the small knowledge of the problem's landscape which we generally dispose of. In addition, these heuristics typically can not guarantee that the SGD iterates start "close enough" to a minimizer, i.e. in the region where the PL condition holds, such that the method enjoys a linear rate of convergence. Therefore, the ideal scenario would be to be able to exploit the stronger local structure while the method's iterates gradually approach a minimizer and independently from the starting point. In this regard, homotopy methods are a general strategy for tackling difficult optimization problems by gradually transforming a simplified version of a target problem, or a version with a known minimizer, back to its original form while following a solution along the way. Consequently, they preserve in each step the vicinity to a minimizer of the currently tackled problem, allowing the solver to always work in regions where the problems exhibit stronger structures. In general terms, homotopy methods (Allgower & Georg, 2003) are a widely and successfully used mathematical tool to efficiently solve various problems in numerical analysis, e.g. (Deuflhard, 2011), (Liao, 2012). Such methods are also suitable to solve complex non-convex optimization problems where no or only little prior knowledge regarding the localization of the solutions is available, allowing for the exploitation of the stronger local structures of the problems in order to achieve fast global convergence, e.g. (Xiao & Zhang, 2012; Lin & Xiao, 2014; Suzumura et al., 2014; Gargiani et al., 2020).

In this work, we propose a stochastic first-order numerical method to solve Problem 1, called Homotopy-Stochastic Gradient Descent (H-SGD), which is based on the combination of the homotopy method and SGD. After introducing the method and discussing the related work (Section 2), our contributions are as follows

1. In Section 3, we provide a general theoretical analysis of H-SGD under some mild assumptions, showing that, if the increments in the homotopy parameter are "small enough", the proposed method tracks in expectation an $r$-optimal solution across homotopy iterations. We then show that, in the same setting, H-SGD can achieve a global linear rate of convergence to a minimizer's neighborhood when used in combination with a specific schedule for the homotopy parameter, i.e. $\Delta\lambda_i$ decreases exponentially across homotopy iterations.

2. In Section 4, we empirically evaluate the performance of H-SGD. Our experiments not only confirm the theoretical results derived in Section 3 but also show that H-SGD with a smartly designed homotopy map can outperform SGD.

## 2 HOMOTOPY-SGD

Homotopy-Stochastic Gradient Descent (H-SGD) is based on the combination of the homotopy method and SGD, in the hope of combining the best of both worlds. In particular, the goal is that of maintaining the advantageous properties of SGD, such as its cheap iterations and fast local convergence under PL condition, while maximally exploiting the stronger local structures via the homotopy scheme. Therefore, H-SGD relies on the definition of a homotopy map $f(w, \lambda) : \mathbb{R}^d \times [0, 1] \to \mathbb{R}$, such that, when $\lambda = 0$ we recover a well-behaved function, e.g. convex, or a function with a known minimizer's localization, and by increasing the $\lambda$ parameter, also called homotopy parameter, we gradually morph it in order to finally end up with our target objective function $f(w, 1) = f(w)$ (see Suciu, 2016, for more details on homotopy functions). By using such a homotopy map, H-SGD finds an approximate solution of Problem 1 by approximately solving a series of parametric problems that gradually leads to the target one. In particular, in each homotopy

iteration $i$, H-SGD tackles a parametric problem of the form

$$w_i^* \in \arg \min_{w \in \mathbb{R}^d} f(w, \lambda_i), \tag{3}$$

where the homotopy parameter $\lambda_i$ is slightly increased at each homotopy iteration. As confirmed by our theoretical analysis, if the variations of the homotopy parameter, i.e. $\Delta\lambda_i$, are "small enough" across homotopy iterations, the method is able to track in expectation an $r$-optimal solution from source to target problem. As shown in Algorithm 1, H-SGD takes as input an approximate solution for the problem associated with $f(w, 0)$, i.e. $w_0$, and is then characterized by two loops: in the outer loop (homotopy iterations) the method defines a new objective function by increasing the homotopy parameter (line 5 in Algorithm 1), and in the inner loop (warm-started SGD iterations) the current homotopy problem is approximately solved with $k$ iterations of SGD starting from the previously derived approximate solution, i.e. $w_{i-1}$ (line 6 in Algorithm 1). Different functions $h : \mathbb{N} \to (0, 1]$ to determine the increment $\Delta\lambda_i$ in the homotopy parameter at each homotopy iteration can be used. As shown in our analysis, this function greatly impacts on the method's properties and convergence rate. In particular, our theoretical analysis confirms that, when a specifically designed scheme for $\Delta\lambda_i$ is deployed, i.e. exponentially decreasing schedule, H-SGD is effective in guaranteeing a global linear rate of convergence to a neighborhood of a minimizer of our target problem, while, given the same setting, vanilla SGD can only ensure a global sublinear rate of convergence.

---

**Algorithm 1** Homotopy-Stochastic Gradient Descent (H-SGD)

---

1: **input:** $w_0 \in \mathbb{R}^d$, $n \in \mathbb{N}$, $k \in \mathbb{N}$, $h : \mathbb{N} \to (0, 1]$ with $\sum_{i=1}^n h(i) = 1$ and $\alpha > 0$
2: **initialization:** $i = 0$, $\lambda_0 = 0$
3: **for** $i = 1, \ldots, n$ **do**
4:     $\Delta\lambda_i \leftarrow h(i)$
5:     $\lambda_i \leftarrow \lambda_{i-1} + \Delta\lambda_i$
6:     $w_i \leftarrow \text{SGD}(w_{i-1}, \alpha, k, f(\cdot, \lambda_i))$
7: **output:** $w_i$

---

## 2.1 RELATED WORK

Finding a solution of Problem 1 when the objective function is non-convex is often quite challenging. Different heuristics hence have been proposed to speed up and improve the optimization of such problems, many of which to be used in combination with stochastic first-order methods such as SGD. In this regard, the proposed method, despite being new in its general formulation and analysis, finds many interesting similarities and connections with existing heuristics in the machine learning literature, e.g. (Bengio et al., 2009; Hinton et al., 2012; Sutskever et al., 2013a). We now briefly discuss some of the state-of-the-art optimization techniques and initialization strategies for solving Problem 1 that are most related to H-SGD, drawing connections with existing and ongoing research works and in the hope that our analysis can also lead to a new interpretation of some widely used techniques which so far lack a more rigorous theoretical description and analysis.

**Graduated Optimization.** The graduated optimization approach (Blake & Zisserman, 1987), also known as *coarse-to-grained optimization method*, is a general heuristic to solve complex non-convex problems that relies on the basic principles of the homotopy method. As the name suggests, at first a coarse-grained and "easy-to-solve" version of the target problem is generated via a smoothing operation. The method then proceeds by gradually refining the problem versions, using the previous solution as initial point. Graduated optimization has been utilized explicitly and implicitly as heuristic in many machine learning and computer vision applications, e.g. object localization (Mobahi et al., 2012), manifold learning (Gashler et al., 2007), optical flow (Brox & Malik, 2011). Unfortunately, many of these techniques have practical and/or theoretical gaps, as they generally lack a rigorous running time and convergence analysis, and/or, as in (Mobahi & Fisher III, 2014) and (Hazan et al., 2016), they rely on an expensive method, i.e. Gaussian smoothing, to construct coarse-grained versions of the original target problem. Regarding theoretical contributions on graduated optimization methods for solving Problem 1, Hazan et al. (2016) are the first and only, to the best of our knowledge, to provide a theoretical analysis for the running time and convergence rate of a graduated optimization method based on an approximate, yet still expensive, type of Gaussian smoothing and SGD. Unfortunately, their analysis shows two major limitations. First, it relies on their

Gaussian smoothing approximation as homotopy map, which limits the generality of the conducted analysis, while our analysis is independent from the specific formulation of the homotopy map used. Second, the analysis is based on the assumption of local strong convexity, which is a quite strong requirement and hence might lead to considerably smaller local regions than those considered in our analysis (Karimi et al., 2017). To conclude this short overview on graduated optimization, many successful optimization heuristics proposed in the machine learning literature are implicitly related to graduated optimization and, consequently, to homotopy methods, such as curriculum learning (Bengio et al., 2009), simulated annealing (Kirkpatrick et al., 1983), noise injection techniques (Hinton et al., 2012), smart initialization (Sutskever et al., 2013a) and layer-wise pretraining (Bengio et al., 2006).

**Transfer Learning.** Due to the massive amount of computational resources required by the development of modern machine learning applications, the community has started to explore the possibility of re-using learned parameters across different tasks, leading to the development of many new transfer-learning algorithms, e.g. (Torrey & Shavlik, 2010; Pan & Yang, 2010; Yosinski et al., 2014; Gargiani et al., 2020). A simple yet often effective way to transfer knowledge across different tasks consists in using warm-start initialization. In this perspective, transfer-learning boils down to a sort of smart initialization heuristic. A first connection between homotopy methods and transfer-learning was underlined in (Gargiani et al., 2020). The authors propose a transfer-learning algorithm based on the homotopy method and SGD via the definition of a homotopy map that transforms a source task into a target task. The method comes together with a general theoretical analysis that is independent from the specific homotopy map adopted and shows that, under some assumptions, the algorithm can track in expectation an approximate solution from source to target task, i.e. optimality-tracking. Unfortunately, the method's analysis is limited as it only considers constant increments of the homotopy parameter, which automatically degrades the linear rate of the local solver to a sublinear one for the homotopy-based method. In addition, as in (Hazan et al., 2016), the analysis relies on the assumption of local strong convexity, which might hold in a significantly smaller neighborhood of the minimizers than the PL condition (Karimi et al., 2017).

## 3 THEORETICAL ANALYSIS

In this section, we provide a general theoretical analysis of H-SGD as described in Algorithm 1. In particular, after discussing the required underlying assumptions (Section 3.1), and the fundamental theoretical preliminaries (Section 3.2), first we analyze the optimality tracking properties of the proposed method (Section 3.3), and then we show that, with a specifically designed scheme for the homotopy parameter, H-SGD enjoys linear convergence to a minimizer's neighborhood (Section 3.4). The analysis we conduct is independent from the specific type of homotopy map adopted and it applies to any scenario where the assumptions listed in Section 3.1 hold.

Recall that the proposed method is based on sequentially and approximately solving a series of $n$ unconstrained parametric problems of the form

$$\arg \min_{w \in \mathbb{R}^d} f(w, \lambda_i), \quad \forall i = 1, \ldots, n, \tag{4}$$

where $\lambda_i < \lambda_{i+1}$, $\lambda_n = 1$, $\lambda_i \in (0, 1]$. In addition, H-SGD relies on the availability of an approximate solution $w_0$ for the source problem with $\lambda_0 = 0$ as starting point. We use $w_i$ to denote the derived approximate solution for the problem associated with parameter $\lambda_i$ that is obtained by applying $k > 0$ iterations of SGD starting from the previously derived approximate solution for the problem with parameter $\lambda_{i-1}$, $\forall i = 1, \ldots, n$.

Notice that $w_{i-1,t} = w_{i-1,t}(\xi_{[i-1,t-1]})$ for $t = 1, \ldots, k$ with $w_{i-1,k} = w_i(\xi_{[i]})$ is used to refer to the random vector generated at the $i$-th homotopy iteration after $t$ iterations of SGD, where $\xi_{[i-1,t-1]} = (w_0, \xi_0^1, \ldots, \xi_{k-1}^1, \ldots, \xi_0^{i-1}, \ldots, \xi_{k-1}^{i-1}, \xi_0^i, \ldots, \xi_{t-1}^i)$ and $\xi_{[i]} = (w_0, \xi_0^1, \ldots, \xi_{k-1}^1, \ldots, \xi_0^{i-1}, \ldots, \xi_{k-1}^{i-1}, \xi_0^i, \ldots, \xi_{k-1}^i)$ with $\xi_{[0]} = w_0$ are used to refer to the collection of all random sources up to the current iteration. We use $U^*(\lambda)$ to denote the set of local (and global) minimizers of the parametric Problem 1.

### 3.1 ASSUMPTIONS

We now list and discuss the assumptions that we consider throughout our analysis. Together with the standard smoothness and bounded variance assumptions, we also introduce three regularity

assumptions, which describe the localization of the solution map and how the objective function $f$ changes by varying the homotopy parameter across iterations. In addition to these assumptions, we also consider a more general and local version of the standard PL condition.Consequently, unlike the settings considered in (Karimi et al., 2016) and (Vaswani et al., 2019), where the standard PL condition is unrealistically required to hold globally, ours is often encountered in many different non-convex scenarios (see Karimi et al., 2017, for more details).

**Assumption 3.1** (existence of a regular localization of the solution map). *Assume there exists a set* $\Omega \subseteq \mathbb{R}^d \times [0,1]^z$ *such that* $W^*(\lambda) := \Omega \cap U^*(\lambda)$ *and* $\Sigma := \left\{ (y, \lambda) \in \mathbb{R}^d \times [0,1]^z \,|\, y \in W^*(\lambda) \right\}$ *are both non-empty and connected. Moreover, we assume that for a given $\lambda$ all the points in $W^*(\lambda)$ are associated with the same objective function value, which we denote as $f^*(\lambda) := f(y, \lambda)$ for all $y \in W^*(\lambda)$.*

Notice that Assumption 3.1 does not imply vector-valued solutions of the parametric Problem 1.

**Assumption 3.2** (regularity 1). *Assume there exists $\delta > 0$ such that*

$$|f(w, \tilde{\lambda}) - f(w, \hat{\lambda})| \leq \delta \|\tilde{\lambda} - \hat{\lambda}\|, \quad \forall w \in \mathbb{R}^d, \forall \tilde{\lambda}, \hat{\lambda} \in [0,1]^z. \tag{5}$$

**Assumption 3.3** (regularity 2). *Assume there exists $\gamma > 0$ such that*

$$|f^*(\tilde{\lambda}) - f^*(\hat{\lambda})| \leq \gamma \|\tilde{\lambda} - \hat{\lambda}\|, \quad \forall \tilde{\lambda}, \hat{\lambda} \in [0,1]^z. \tag{6}$$

**Assumption 3.4** ($L$-smoothness). *Assume there exists $L > 0$ such that*

$$\|\nabla_w f(\tilde{w}, \lambda) - \nabla_w f(\hat{w}, \lambda)\| \leq L \|\tilde{w} - \hat{w}\|, \quad \forall \tilde{w}, \hat{w} \in \mathbb{R}^d, \forall \lambda \in [0,1]^z. \tag{7}$$

See Remark C.1 in Section C of the Appendix for more details on Assumptions 3.2 and 3.3.

**Assumption 3.5** (bounded "variance"). *Consider $f(w, \lambda)$ with $\lambda \in [0,1]^z$ and let $g(w, \xi, \lambda)$ be the stochastic estimate of the true gradient $\nabla_w f(w, \lambda)$ used in SGD with noise $\xi$. Assume that*

$$\mathrm{E}_\xi \left[ g(w, \xi, \lambda) \right] = \nabla_w f(w, \lambda), \quad \forall w \in \mathbb{R}^d, \forall \lambda \in [0,1]^z. \tag{8}$$

*and that there exists $\sigma^2 \geq 0$ such that*

$$\mathrm{E}_\xi \left[ \|g(w, \xi, \lambda) - \nabla_w f(w, \lambda)\|^2 \right] \leq \sigma^2, \quad \forall w \in \mathbb{R}^d, \forall \lambda \in [0,1]^z. \tag{9}$$

**Assumption 3.6** ("expected" PL condition). *Consider $f(w, \lambda_i)$ with $\lambda_i \in [0,1]^z$ and let $w_{i-1,t} = w_{i-1,t}(\xi_{[i-1,t-1]})$ denote the iterate that is obtained at the $i$-th homotopy iteration after $t$ iterations of SGD with $t \leq k$. Assume that there exist $B > \frac{\sigma^2}{2\mu}$ and $\mu > 0$ such that, if $\mathrm{E}_{\xi_{[i-1,t-1]}} \left[ f(w_{i-1,t}, \lambda_i) \right] - f^*(\lambda_i) \leq B$, then*

$$\mathrm{E}_{\xi_{[i-1,t-1]}} \left[ \|\nabla_w f(w_{i-1,t}, \lambda_i)\|^2 \right] \geq 2\mu \cdot \left[ \mathrm{E}_{\xi_{[i-1,t-1]}} \left[ f(w_{i-1,t}, \lambda_i) \right] - f^*(\lambda_i) \right]. \tag{10}$$

See Remark C.2 for additional details on Assumption 3.6.

## 3.2 FUNDAMENTAL THEORETICAL PRELIMINARIES

Before proceeding with the main theoretical contributions, we revise and adjust the existing results in the literature on global error bounds of SGD, i.e. (Vaswani et al., 2019), to also hold in the considered setting. The extended results are then used for the derivations in Section 3.3 and 3.4.

**Proposition 3.7.** *Consider $f(w, \lambda_i)$ with $\lambda_i \in [0,1]^z$ and let $w_{i-1,t} = w_{i-1,t}(\xi_{[i-1,t-1]})$ denote the iterate obtained at the $i$-th homotopy iteration by applying $t$ iterations of SGD with $t \leq k - 1$ and $\alpha \leq \frac{1}{L}$. Under Assumptions 3.1 and 3.4-3.6, if $\mathrm{E}_{\xi_{[i-1,t-1]}} \left[ f(w_{i-1,t}, \lambda_i) \right] - f^*(\lambda_i) \leq B$, then $\mathrm{E}_{\xi_{[i-1,t]}} \left[ f(w_{i-1,t+1}, \lambda_i) \right] - f^*(\lambda_i) \leq B$.*

*Proof.* See Section D in the Appendix for a proof. □

**Theorem 3.8.** *Consider the minimization of $f(w, \lambda_i)$ with $\lambda_i \in [0,1]^z$ via SGD. Let $w_{i-1} = w_{i-1}(\xi_{[i-1]})$ be the random initial point associated with the $i$-th homotopy iteration with*

$\mathbb{E}_{\xi_{[i-1]}}[f(w_{i-1}, \lambda_i)] - f^*(\lambda_i) \leq B$ and $w_{i-1,t} = w_{i-1,t}(\xi_{[i-1,t-1]})$ denote the $t$-th SGD iterate with $t \leq k$. Under Assumptions 3.1 and 3.4-3.6, SGD with a constant step-size $\alpha \leq \frac{1}{L}$ attains the following convergence rate to a minimizer's neighborhood

$$\mathrm{E}_{\xi_{[i-1,t-1]}}[f(w_{i-1,t}, \lambda_i) - f^*(\lambda_i)] \leq \rho^t \mathrm{E}_{\xi_{[i-1]}}[f(w_{i-1}, \lambda_i) - f^*(\lambda_i)] + \frac{\sigma^2}{2\mu}, \qquad (11)$$

with $\rho := (1 - \alpha\mu)$. With $\alpha = \frac{1}{L}$, we obtain $\rho = \left(1 - \frac{\mu}{L}\right)$.

*Proof.* See Section E in the Appendix for a proof. □

### 3.3 OPTIMALITY TRACKING

In the following, we define the function $\phi_v(\lambda_i) := \mathrm{E}_v[f(v, \lambda_i)] - f^*(\lambda_i)$ where $v$ is $d$-dimensional real random vector. As in (Gargiani et al., 2020) but in a more relaxed setting, i.e. local PL in place of local strong-convexity, we study the optimality tracking properties of H-SGD. In particular, under the considered assumptions and by exploiting the previously introduced results on the convergence of SGD, with Theorem 3.10 we characterize the maximum allowed variation of the homotopy parameter across homotopy iterations of H-SGD such that, if $\phi_{w_i}(\lambda_i) \leq r$, then also $\phi_{w_{i+1}}(\lambda_{i+1}) \leq r$. The upper bound that we derive depends on the number of iterations $k$ performed with SGD as well as on the convergence characteristics of SGD and the structural properties of the parametric problems. This result applied recursively across homotopy iterations leads to conclude that, if we adopt a "small enough" increasing step for the homotopy parameter, H-SGD can track in expectation an $r$-optimal solution from source to target problem.

Before proceeding with the actual optimality tracking analysis (Theorem 3.10), we study the conditions on $w_i$ and $\Delta\lambda_{i+1}$ such that $\phi_{w_i}(\lambda_{i+1}) \leq B$, where $w_i$ is the approximate solution of the problem associated with parameter $\lambda_i$ that is also used as starting point for the next parametric problem.

**Lemma 3.9.** *Assume* $\|\lambda_{i+1} - \lambda_i\| \leq \epsilon$, $0 \leq \epsilon < \frac{B}{\delta+\gamma}$ *and let* $w_i$ *denote the $i$-th iterate of Algorithm 1 with* $\alpha \leq \frac{1}{L}$. *Under Assumptions 3.1- 3.3 and 3.4- 3.6, if* $\phi_{w_i}(\lambda_i) \leq B - (\delta + \gamma)\epsilon$, *then* $\phi_{w_i}(\lambda_{i+1}) \leq B$. *In addition, let* $k_{\max} := \left\lceil \log_\rho \left(1 - \frac{2\mu(\delta+\gamma)\epsilon+\sigma^2}{2\mu B}\right) \right\rceil$. *If* $\phi_{w_i}(\lambda_{i+1}) \leq B$, $0 \leq \epsilon < \frac{1}{\delta+\gamma}\left(B - \frac{\sigma^2}{2\mu}\right)$ *and* $k \geq k_{\max}$, *then* $\phi_{w_{i+1}}(\lambda_{i+1}) \leq B - (\delta + \gamma)\epsilon$.

*Proof.* See Section F in the Appendix for a proof. □

**Theorem 3.10.** *Assume there exists* $\frac{\sigma^2}{2\mu} < r \leq B$ *and* $\tilde{\epsilon} := \min\{\epsilon_1, \epsilon_2\}$ *with*

$$\epsilon_1 := \frac{1}{(\delta+\gamma)}(B-r), \quad \epsilon_2 := \frac{(1-\rho^k)r - \sigma^2/2\mu}{\rho^k(\delta+\gamma)}. \qquad (12)$$

*In addition, let* $k_{\max} := \left\lceil \log_\rho \left(1 - \frac{\sigma^2}{2\mu r}\right) \right\rceil$. *Consider Algorithm 1 with* $\alpha \leq \frac{1}{L}$, $k \geq k_{\max}$ *and* $\|\lambda_{i+1} - \lambda_i\| \leq \epsilon$, *where* $0 \leq \epsilon \leq \tilde{\epsilon}$. *Under Assumptions 3.1- 3.3 and 3.4- 3.6, if* $\phi_{w_i}(\lambda_i) \leq r$, *then* $\phi_{w_{i+1}}(\lambda_{i+1}) \leq r$.

*Proof.* See Section G in the Appendix for a proof. □

See Figure 2 in Section B of the Appendix for a graphical representation of the derived results.

### 3.4 LINEAR CONVERGENCE RATE

We now study the convergence rate of H-SGD and, in particular, if it is possible to recover a global linear rate of convergence to a minimizer's neighborhood. The results derived in Theorem 3.11 confirm that, in the considered setting and with a specifically designed schedule for the homotopy parameter, H-SGD achieves the desired rate of convergence.

**Theorem 3.11.** *Let $\tilde{\rho} \in \left(1 - \frac{\sigma^2}{2\mu} \frac{1}{B}, 1\right)$ and consider Algorithm 1 with $\alpha \leq \frac{1}{L}$, $\phi_{w_0}(\lambda_0) \leq r$ with $\frac{\sigma^2}{2\mu} \frac{1}{(1-\tilde{\rho})} \leq r \leq B$ and $k \geq \log_\rho(\tilde{\rho})$. In addition, let $\epsilon_1 := \frac{1}{(\delta+\gamma)}(B-r)$ and*

$$C_{\tilde{\rho}} := \begin{cases} 1 & \text{if } k \geq \log_\rho(\tilde{\rho}) - \log_\rho\left(1 + \frac{\delta+\gamma}{\varepsilon_0}\right) \\ \frac{\tilde{\rho}-\rho^k}{\rho^k} \frac{\varepsilon_0}{(\delta+\gamma)} & \text{otherwise,} \end{cases} \tag{13}$$

*with $\varepsilon_0 := \mathrm{E}_{\xi_{[0]}}\left[f(w_0, \lambda_0)\right] - f^*(\lambda_0)$. Under Assumptions 3.1- 3.3 and 3.4- 3.6, if $\|\lambda_{i+1} - \lambda_i\| \leq \min\left\{e^{-\eta i}, \epsilon_1\right\}$ with $\eta \geq \ln\left(C_{\tilde{\rho}}\, \tilde{\rho}\right)$, then*

$$\mathrm{E}_{\xi_{[i+1]}}\left[f(w_{i+1}, \lambda_{i+1})\right] - f^*(\lambda_{i+1}) \leq \tilde{\rho}^{i+1}\left[\mathrm{E}_{\xi_{[0]}}\left[f(w_0, \lambda_0)\right] - f^*(\lambda_0)\right] + \frac{\sigma^2}{2\mu}\sum_{j=0}^{i}\tilde{\rho}^j. \tag{14}$$

*Proof.* See Section H in the Appendix for a proof. □

## 4 EXPERIMENTAL EVALUATION

In this section, we empirically validate the theoretical results derived in Theorem 3.11. First, we consider a 1-dimensional toy regression problem to illustrate and visualize some of the basic properties of H-SGD. In this easy scenario, the introduced assumptions can be trivially verified by inspection (see the Figures 3- 4 in the Appendix). We then move to more complex and high-dimensional scenarios where the assumptions can not be verified. Inspired by Finn et al. (2017) and Gargiani et al. (2020), we consider the task of regressing with a neural network from input to output of a sinusoidal wave. Finally, we also consider a non-convex classification task based on the combination of logistic regression with a non-linear model for moon-shaped binary data (Chapelle et al., 2006). In all the considered scenarios, we compare the numerical performance of H-SGD with those of SGD and tune the step-size based on the performance of the latter.

### 4.1 TOY-PROBLEM

We start with an easy regression problem motivated by Mobahi (2016): a 1-dimensional neural network with erf as activation function (see Figure 9 in Section B of the Appendix for a graphical representation). We generate a synthetic dataset of $N = 100$ samples, where $x_j \in [-1, 1]$, $y_j = 3 \cdot x_j + \epsilon_j$ and $\epsilon_j \sim \mathcal{N}(0, 1)$. Regarding the choice of a value for the step-size, we use an estimate $\tilde{L}$ of the smoothness constant $L$ and set $\alpha = 1/\tilde{L}$. As loss, we use the mean squared error, which, composed with the regressing model, leads to the following non-convex optimization problem

$$w^* = \arg\min_{w \in \mathbb{R}} \frac{1}{N}\sum_{j=1}^{N}(y_j - \mathrm{erf}(w \cdot x_j))^2. \tag{15}$$

By plotting the objective function with respect to $w$ (see Fig. 3 in Section B of the Appendix), it is easy to observe that the PL condition holds globally, with the value of $\mu$ increasing by approaching the minimizer and $\mu \to 0$ for $w \to \pm\infty$ (see Fig. 4 in the Appendix). Consequently, SGD enjoys a global linear convergence rate as proved in (Vaswani et al., 2019) but the rate itself, which depends on the value of $\mu$, will dramatically worsen the further the iterates are from the minimizer, leading to a great overall sensitivity of the method in terms of convergence rate to the initialization. On the other side, H-SGD, thanks to the homotopy principle, goes around that issue by preserving the vicinity to the minimizer of the current homotopy problem at each homotopy iteration (see Figures 7 and 8 in Section B of the Appendix). In order to achieve that, given $w_0$ as initial value, we set $y_{j,0} = w_0 \cdot x_j$ for all $j = 1, \ldots, N$ and define the following homotopy transformation

$$y_{j,\lambda} = \lambda y_j + (1 - \lambda) y_{j,0}. \tag{16}$$

As suggested by the theory (see Theorem 3.11), we select an exponentially decreasing scheme for the increment $\Delta\lambda_i$ in order to achieve linear convergence. As shown in Figure 5 in the Appendix, both H-SGD and SGD enjoy a linear rate of convergence, but H-SGD shows a superior numerical performance. This is due to the fact that the method is designed such that its iterates always lie in the neighborhood of the minimizers where more favorable values of $\mu$ lead to a faster convergence. This fact allows H-SGD to enjoy a faster global convergence rate than that of SGD.

## 4.2 Regression with Deep Neural Networks

Our second experiment is inspired by Finn et al. (2017) and Gargiani et al. (2020) and focuses on studying the numerical performance of H-SGD on the task of regressing from input to output of a sinusoidal function corrupted by Gaussian noise. In particular, the input data are sampled uniformly from the interval $[-1, 1]$ and $y_j = \sin(10 \cdot x_j) + \epsilon_j$ with $\epsilon_j \sim \mathcal{N}(0, 0.1)$ for all $j = 1, \ldots, 500$. The regressor is a feedforward neural network with two hidden layers, each of 10 units, and hyperbolic tangent as activation function. As for the previous benchmark, we use the mean squared error as loss. We employ the same values of step-size $\alpha$ and mini-batch $M$ for both H-SGD and SGD, where the step-size value is tuned based on the numerical performance of SGD for the selected mini-batch size ($M = 5$). Regarding H-SGD, we set $y_{j,0} = x_j^2 + \epsilon_j$ with $\epsilon_j \sim \mathcal{N}(0, 0.01)$ and employ the same homotopy mapping as in Equation 16. As shown in Figure 6 in the Appendix, also in this scenario H-SGD shows a superior numerical performance than SGD, i.e. H-SGD reaches a loss of $10^{-1}$ roughly 4 times faster and achieves convergence more than 2 times faster than SGD. The superior numerical performance of H-SGD can be attributed to its ability of tracking a solution across homotopy iterations (see Figure 10 in the Appendix) which ensures the method to always work in the vicinity of a minimizer.

## 4.3 Non-Linear Binary Classification with Logistic Regression

Finally, we test H-SGD also on a classification benchmark with a logistic regression task. In particular, we use a 2-dimensional binary moon-shaped dataset (Chapelle et al., 2006) with $1000$ samples corrupted by Gaussian noise. As the dataset is clearly not linearly separable, we opt for a cubic model, which, used in combination with the logistic regression framework, leads to a non-convex objective function where the optimization variable $w$ is the collection of the model's coefficients. For both H-SGD and SGD we use a mini-batch size of $20$ and tune the value of the step-size on the SGD's performance for that mini-batch size. Regarding H-SGD, we use as source task the one obtained considering a linear instead of a cubic model, which results in a convex and hence "easy" optimization problem. We then gradually increase the non-linearity of the model, i.e. non-convexity of the problem, until reaching in the final homotopy iteration the target problem with the desired cubic model. This homotopy map is obtained by multiplying the coefficients of the non-linear terms in the model by $\lambda$ as follows

$$\lambda(c_1 x_{j,1}^3 + c_2 x_{j,2}^3 + c_3 x_{j,1}^2 + c_4 x_{j,2}^2 + c_5 x_{j,1}^2 x_{j,2} + c_6 x_{j,1} x_{j,2}^2) + c_7 x_{j,1} + c_8 x_{j,2} + c_9 \,. \tag{17}$$

For the homotopy parameter we adopt the increasing schedule that is suggested in Theorem 3.11. H-SGD outperforms SGD by reaching an error of $0.1$ more than two times faster than SGD (see Figure 11 in the Appendix).

## 5 Conclusions and Future Work

In this paper we propose a new first-order stochastic method for non-convex large-scale problems, called Homotopy-SGD (H-SGD), based on the combination of homotopy methods and SGD. This new homotopy-based optimization method allows one to exploit easy-to-solve or already-solved problems to solve new and complex ones. This is achieved by approximately and sequentially solving a sequence of optimization problems where the source problem is gradually morphed via a homotopy map into the target one. We conduct a theoretical analysis of the optimality tracking properties and convergence rate of H-SGD under some realistic and mild assumptions. The theoretical results are confirmed by some empirical evaluations, which also show the great potential in terms of performance of combining SGD with an homotopy strategy. In addition, H-SGD shows interesting connections with many practical existing heuristics proposed in the machine learning literature to speed up the convergence of first-order methods, allowing for a new and more rigorous interpretation of the latter. The current major limitation of the method relies in the design of the homotopy map. Future work should focus on exploiting the specific problem structure to design optimal homotopy maps. Moreover, under additional assumptions, more theoretical results concerning the quality of the tracked solutions could be derived.

ACKNOWLEDGMENTS

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

# A  APPENDIX

# B  ADDITIONAL FIGURES

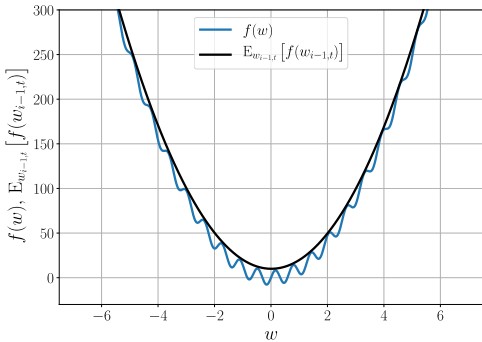

Figure 1: Graphical representation of a 1-dimensional function with $w_{i-1,t} \sim \mathcal{N}(0, 1)$ that satisfies the "expected" PL condition (Assumption 3.6) but not the classical PL condition in the considered region.

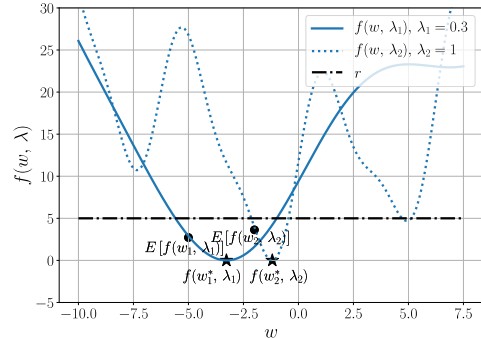

Figure 2: Graphical representation of the results derived in Theorem 3.10 on the optimality tracking properties of H-SGD for a general non-convex 1-dimensional function. In particular, as shown in the figure, under the considered assumptions and for "small enough" variations of the homotpy parameter, H-SGD tracks in expectation an $r$-optimal solution across homotopy iterations.

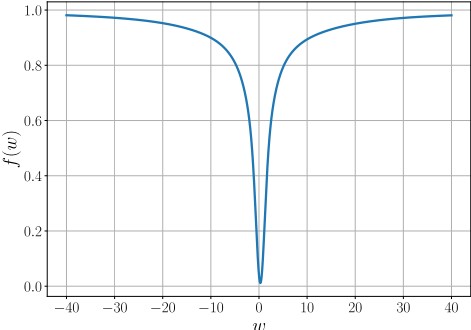

Figure 3: Graphical representation of the objective function in Problem equation 15 vs $w$.

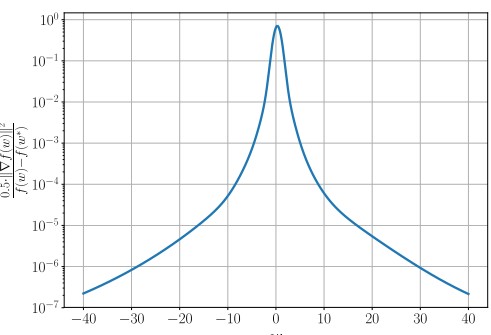

Figure 4: Visualization of the estimated $\mu$ parameter for the objective function in Problem equation 15 vs $w$.

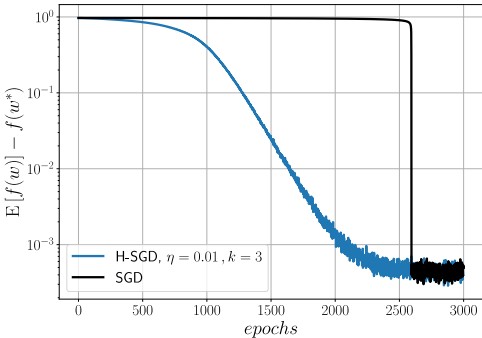

Figure 5: Expected optimality gap of H-SGD (blue) and SGD (black) averaged across 100 runs vs epochs for the toy-case described in Section 4.1

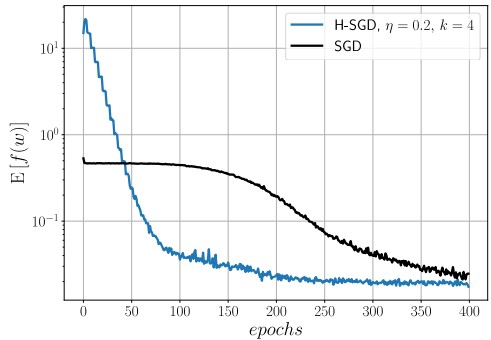

Figure 6: Expected loss of H-SGD (blue) and SGD (black) averaged across 100 runs vs epochs for the sine-wave regression case described in Section 4.2

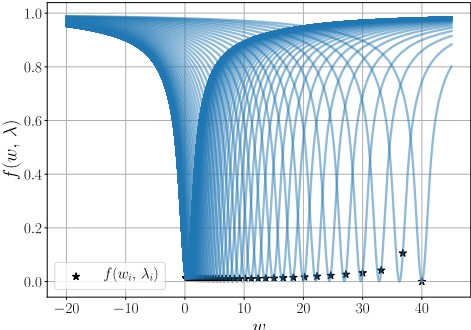

Figure 7: Visualization of the homotopy objective functions vs $w$ for different values of homotopy parameter. The black stars represent $(w_i, f(w_i, \lambda_i))$, i.e. the H-SGD iterates with associated objective function value.

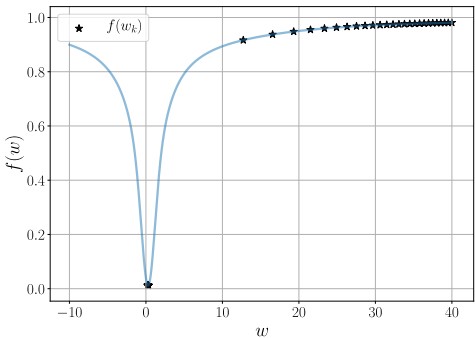

Figure 8: Visualization of the target objective functions vs $w$. The black stars represent $(w_k, f(w_k))$, i.e. the SGD iterates with associated objective function value.

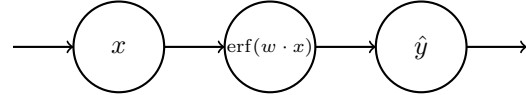

Figure 9: Graphical representation of the 1-dimensional neural network deployed for the toy-case experiment described in Section 4.1.

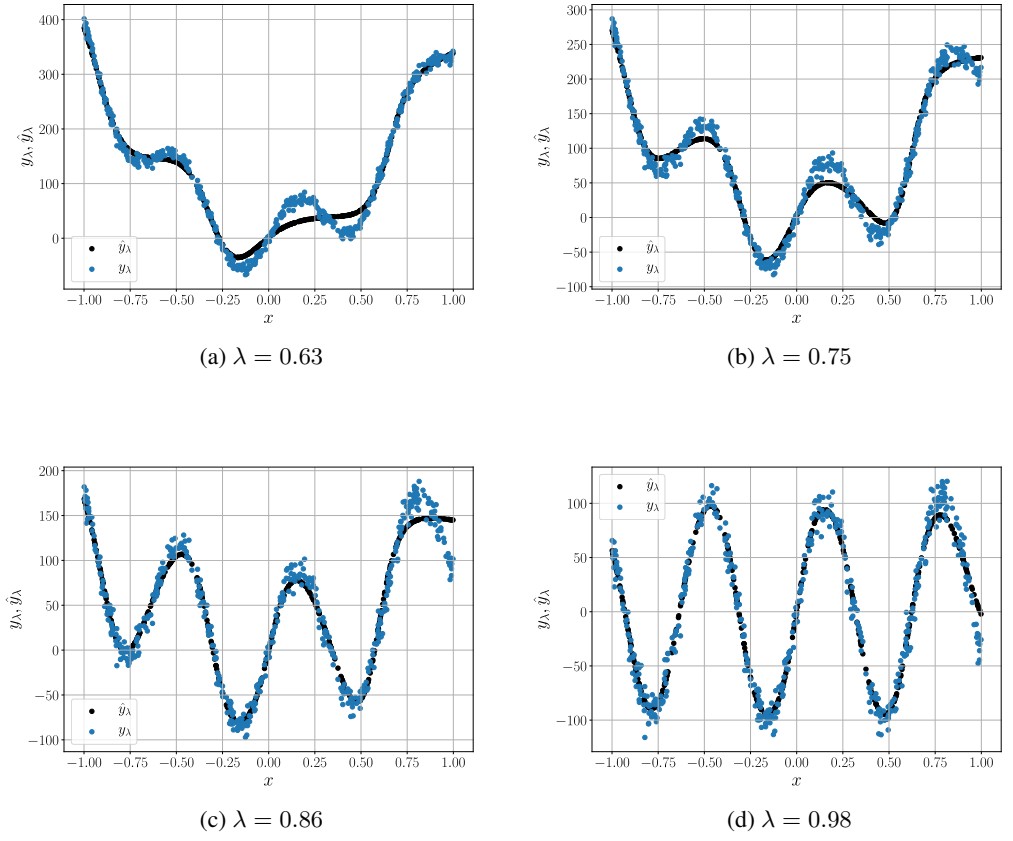

Figure 10: Predicted values $\hat{y}_\lambda$ (blue) vs true values $y_\lambda$ (black) for different values of $\lambda$ generated by using H-SGD.

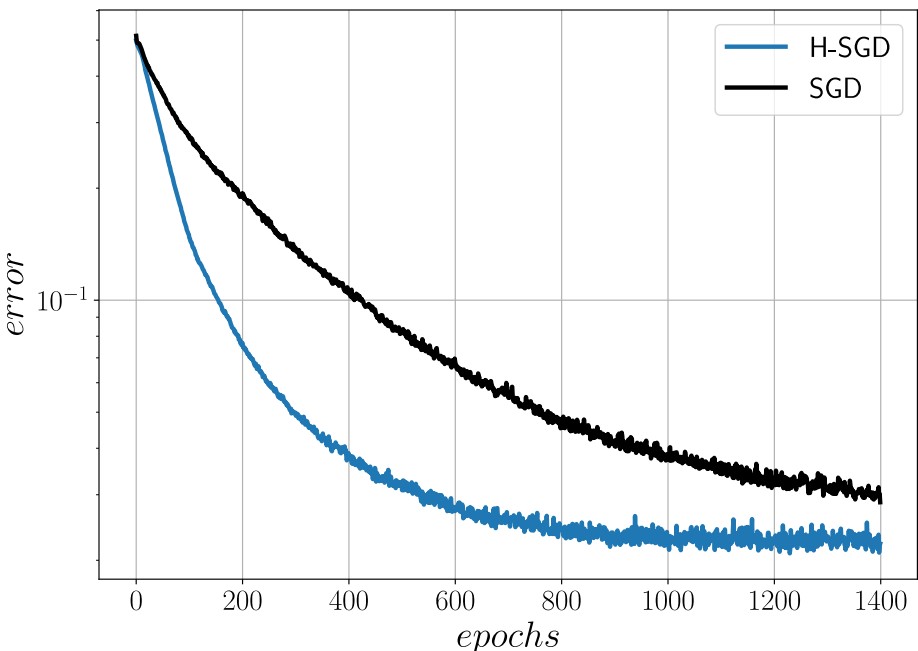

Figure 11: Error averaged across 100 runs of H-SGD (blue) and SGD (black) vs epochs for the classification task described in Section 4.3.

## C   ADDITIONAL REMARKS

**Remark C.1.** *Assumption 3.2 can be read as a local Lipschitz continuity of $f$ with respect to its second argument. Regarding the second regularity assumption (Assumption 3.3), equation 6 relates the variation of the optimal objective function value for the subset of solutions introduced in Assumption 3.1 across homotopy iterations with the variations of the homotopy parameter. When the solution localization of the solution map is vector-valued, i.e. $w^*(\lambda) \equiv W^*(\lambda)$ (this is the case for instance when the Hessian of the objective function is positive definite at those points), this assumption can also be derived directly by combining Assumption 3.2 with the following Lipschitz continuity requirements:*

- *Let $\kappa_1 > 0$, we assume that*

$$|f(w^*(\tilde{\lambda}), \hat{\lambda}) - f(w^*(\hat{\lambda}), \hat{\lambda})| \leq \kappa_1 \|w^*(\tilde{\lambda}) - w^*(\hat{\lambda})\|, \quad \forall \tilde{\lambda}, \hat{\lambda} \in [0, 1]^z. \tag{18}$$

- *Let $\kappa_2 > 0$, we assume that*

$$\|w^*(\tilde{\lambda}) - w^*(\hat{\lambda})\| \leq \kappa_2 \|\tilde{\lambda} - \hat{\lambda}\|, \quad \forall \tilde{\lambda}, \hat{\lambda} \in [0, 1]^z. \tag{19}$$

*In particular, by combining Inequalities equation 18 and equation 19, we obtain*

$$|f(w^*(\tilde{\lambda}), \hat{\lambda}) - f(w^*(\hat{\lambda}), \hat{\lambda})| \leq \kappa_2 \, \kappa_1 \|\tilde{\lambda} - \hat{\lambda}\|, \quad \forall \tilde{\lambda}, \hat{\lambda} \in [0, 1]^z. \tag{20}$$

*To recover Inequality equation 6 where $\gamma := \delta + \kappa_1 \, \kappa_2$, we use Inequalities equation 5 and equation 20 together with the triangle inequality as follows*

$$\begin{aligned}
|f(w^*(\tilde{\lambda}), \tilde{\lambda}) - f(w^*(\hat{\lambda}), \hat{\lambda})| &= |f(w^*(\tilde{\lambda}), \tilde{\lambda}) - f(w^*(\tilde{\lambda}), \hat{\lambda}) + f(w^*(\tilde{\lambda}), \hat{\lambda}) - f(w^*(\hat{\lambda}), \hat{\lambda})| \\
&\leq |f(w^*(\tilde{\lambda}), \tilde{\lambda}) - f(w^*(\tilde{\lambda}), \hat{\lambda})| + |f(w^*(\tilde{\lambda}), \hat{\lambda}) - f(w^*(\hat{\lambda}), \hat{\lambda})| \\
&\leq (\delta + \kappa_1 \kappa_2)\|\tilde{\lambda} - \hat{\lambda}\|.
\end{aligned}$$

$$\tag{21}$$

*Notice that the Assumption in equation 18 is a less restrictive condition of the following general Lipschitz continuity requirement*

$$|f(v, \hat{\lambda}) - f(w, \hat{\lambda})| \leq C\|v - w\|, \quad \forall v, w \in \mathbb{R}^d, \ \forall \hat{\lambda} \in [0, 1]^z, \tag{22}$$

*where $C > 0$.*

**Remark C.2.** *Assumption 3.6 is a more general version of the classical PL condition (see Section 2 Karimi et al., 2016, for more details on the classical PL condition). In particular, it is straightforward to observe that the classical PL condition implies Assumption 3.6, but not vice versa. See Figure 1 in Section B of the Appendix for a graphical representation of a one dimensional example with $w_{i-1,t} \sim \mathcal{N}(0, 1)$ where Assumption 3.6 holds, while the classical PL condition does not. The expected value operates a smoothing of the function landscape resulting in convexity, while the original function shows many bumps that make it non-convex.*

## D  PROOF OF PROPOSITION 3.7

**Proposition D.1.** *Consider $f(w, \lambda_i)$ with $\lambda_i \in [0, 1]^z$ and let $w_{i-1,t} = w_{i-1,t}(\xi_{[i-1,t-1]})$ denote the iterate obtained at the $i$-th homotopy iteration by applying $t$ iterations of SGD with $t \leq k - 1$ and $\alpha \leq \frac{1}{L}$. Under Assumptions 3.4, 3.5 and 3.6, if $E_{\xi_{[i-1,t-1]}}[f(w_{i-1,t}, \lambda_i)] - f^*(\lambda_i) \leq B$, then $E_{\xi_{[i-1,t]}}[f(w_{i-1,t+1}, \lambda_i)] - f^*(\lambda_i) \leq B$.*

*Proof.* For ease of notation, we use the shorthands $w_{i-1,t} = w_{i-1,t}(\xi_{[i-1,t-1]})$ and $g_t = g(w_{i-1,t}, \xi_t^i, \lambda_i)$.

Considering Assumption 3.4 together with the definition of SGD iterate and using Assumptions 3.5 and 3.6, we obtain the following inequalities

$$E_{\xi_{[i-1,t]}}[f(w_{i-1,t+1}, \lambda_i)] \leq E_{\xi_{[i-1,t]}}\left[f(w_{i-1,t}, \lambda_i) - \alpha\langle\nabla f(w_{i-1,t}, \lambda_i), g_t\rangle + \frac{L\alpha^2}{2}\|g_t\|^2\right]$$

$$\overset{\text{Law of Iterated Exp.}}{=} E_{\xi_{[i-1,t-1]}}\left[E_{\xi_t^i}\left[f(w_{i-1,t}, \lambda_i) - \alpha\langle\nabla f(w_{i-1,t}, \lambda_i), g_t\rangle + \frac{L\alpha^2}{2}\|g_t\|^2\Big|\xi_{[i-1,t-1]}\right]\right]$$

$$\overset{\text{Assumption 3.5}}{\leq} E_{\xi_{[i-1,t-1]}}\left[f(w_{i-1,t}, \lambda_i) + \left(-\alpha + \frac{L\alpha^2}{2}\right)\|\nabla f(w_{i-1,t}, \lambda_i)\|^2\right] + \frac{L\alpha^2\sigma^2}{2}.$$

If $\alpha \leq \frac{1}{L}$, then

$$E_{\xi_{[i-1,t]}}[f(w_{i-1,t+1}, \lambda_i)] \leq E_{\xi_{[i-1,t-1]}}\left[f(w_{i-1,t}, \lambda_i) + \left(-\frac{\alpha}{2}\right)\|\nabla f(w_{i-1,t}\lambda_i)\|^2\right] + \frac{L\alpha^2\sigma^2}{2}. \tag{23}$$

We now make use of the "expected" PL condition and derive the following inequalities

$$E_{\xi_{[i-1,t]}}[f(w_{i-1,t+1}, \lambda_i) - f(w_{i-1,t}, \lambda_i)] \leq -\frac{\alpha}{2}E_{\xi_{[i-1,t-1]}}\left[\|\nabla f(w_{i-1,t}, \lambda_i)\|^2\right] + \frac{L\alpha^2\sigma^2}{2}$$

$$\overset{\text{Assumption 3.6}}{\leq} -\alpha\mu\left[E_{\xi_{[i-1,t-1]}}[f(w_{i-1,t}, \lambda_i)] - f^*(\lambda_i)\right] + \frac{L\alpha^2\sigma^2}{2}. \tag{24}$$

From Inequality equation 24 it follows that, whenever $E_{\xi_{[i-1,t-1]}}[f(w_{i-1,t}, \lambda_i)] - f^*(\lambda_i) \geq \frac{\sigma^2}{2\mu}$, the objective function decreases in expectation, i.e. $E_{\xi_{[i-1,t]}}[f(w_{i-1,t+1}, \lambda_i) - f(w_{i-1,t}, \lambda_i)] \leq 0$. Given that by assumption $B > \frac{\sigma^2}{2\mu}$, we can consequently conclude that, if $E_{\xi_{[i-1,t-1]}}[f(w_{i-1,t}, \lambda_i)] - f(w_i^*, \lambda_i) \leq B$, then $E_{\xi_{[i-1,t]}}[f(w_{i-1,t+1}, \lambda_i)] - f^*(\lambda_i) \leq B$. $\square$

## E  PROOF OF THEOREM 3.8

**Theorem E.1.** *Consider the minimization of $f(w, \lambda_i)$ with $\lambda_i \in [0, 1]^z$ via SGD. Let $w_{i-1} = w_{i-1}(\xi_{[i-1]})$ be the random initial point associated with the $i$-th homotopy iteration with*

$\mathbb{E}_{\xi_{[i-1]}}[f(w_{i-1}, \lambda_i)] - f^*(\lambda_i) \leq B$ and $w_{i-1,t} = w_{i-1,t}(\xi_{[i-1,t-1]})$ denote the $t$-th SGD iter­ate with $t \leq k$. Under Assumptions 3.4, 3.5 and 3.6, SGD with a constant step-size $\alpha \leq \frac{1}{L}$ attains the following convergence rate to a minimizer's neighborhood

$$\mathrm{E}_{\xi_{[i-1,t-1]}}[f(w_{i-1,t}, \lambda_i) - f^*(\lambda_i)] \leq \rho^t \mathrm{E}_{\xi_{[i-1]}}[f(w_{i-1}, \lambda_i) - f^*(\lambda_i)] + \frac{\sigma^2}{2\mu}, \qquad (25)$$

with $\rho := (1 - \alpha\mu)$. With $\alpha = \frac{1}{L}$, we obtain $\rho = \left(1 - \frac{\mu}{L}\right)$.

*Proof.* For ease of notation, we use the shorthands $w_{i-1,t} = w_{i-1,t}(\xi_{[i-1,t-1]})$ and $g_t = g(w_{i-1,t}, \xi_t^i, \lambda_i)$.

By combining Assumption 3.4 with the definition of SGD iterate, for all $t = 1, \ldots, k-1$, we obtain the following bound on $f(w_{t+1}, \lambda_i)$

$$f(w_{i-1,t+1}, \lambda_i) \leq f(w_{i-1,t}, \lambda_i) + \langle \nabla f(w_{i-1,t}, \lambda_i), w_{i-1,t+1} - w_{i-1,t}\rangle + \frac{L}{2}\|w_{i-1,t+1} - w_{i-1,t}\|^2$$

$$\overset{w_{i-1,t+1} := w_{i-1,t} - \alpha g_t}{=} f(w_{i-1,t}, \lambda_i) - \alpha\langle \nabla f(w_{i-1,t}, \lambda_i), g_t\rangle + \frac{L\alpha^2}{2}\|g_t\|^2.$$

We now take the expectation with respect to all the sources of randomness involved and then we apply the law of iterated expectations, together with Assumption 3.5

$$\mathrm{E}_{\xi_{[i-1,t]}}[f(w_{i-1,t+1}, \lambda_i)] \leq \mathrm{E}_{\xi_{[i-1,t]}}\left[f(w_{i-1,t}, \lambda_i) - \alpha\langle \nabla f(w_{i-1,t}, \lambda_i), g_t\rangle + \frac{L\alpha^2}{2}\|g_t\|^2\right]$$

$$\overset{\text{Law of Iterated Exp.}}{=} \mathrm{E}_{\xi_{[i-1,t-1]}}\left[\mathrm{E}_{\xi_t^i}\left[f(w_{i-1,t}, \lambda_i) - \alpha\langle \nabla f(w_{i-1,t}, \lambda_i), g_t\rangle + \frac{L\alpha^2}{2}\|g_t\|^2\Big|\xi_{[i-1,t-1]}\right]\right]$$

$$\overset{\text{Assumption 3.5}}{\leq} \mathrm{E}_{\xi_{[i-1,t-1]}}\left[f(w_{i-1,t}, \lambda_i) + \left(-\alpha + \frac{L\alpha^2}{2}\right)\|\nabla f(w_{i-1,t}, \lambda_i)\|^2\right] + \frac{L\alpha^2\sigma^2}{2}.$$

If $\alpha \leq \frac{1}{L}$, then

$$\mathrm{E}_{\xi_{[i-1,t]}}[f(w_{i-1,t+1}, \lambda_i)] \leq \mathrm{E}_{\xi_{[i-1,t-1]}}\left[f(w_{i-1,t}, \lambda_i) + \left(-\frac{\alpha}{2}\right)\|\nabla f(w_{i-1,t}, \lambda_i)\|^2\right] + \frac{L\alpha^2\sigma^2}{2}. \tag{26}$$

We now apply the PL condition to Inequality equation 26, and we obtain

$$\mathrm{E}_{\xi_{[i-1,t]}}[f(w_{i-1,t+1}, \lambda_i)] \overset{\text{Assumption 3.6}}{\leq} \mathrm{E}_{\xi_{[i-1,t-1]}}[f(w_{i-1,t}, \lambda_i) \\ -\alpha\mu(f(w_{i-1,t}, \lambda_i) - f^*(\lambda_i))] + \frac{L\alpha^2\sigma^2}{2}. \tag{27}$$

By subtracting $f^*(\lambda_i)$ on both sides and setting $\alpha = \frac{1}{L}$, we obtain the following inequality

$$\mathrm{E}_{\xi_{[i-1,t]}}[f(w_{i-1,t+1}, \lambda_i) - f^*(\lambda_i)] \leq \left(1 - \frac{\mu}{L}\right)\mathrm{E}_{\xi_{[i-1,t-1]}}[f(w_{i-1,t}, \lambda_i) - f^*(\lambda_i)] + \frac{\sigma^2}{2L}. \tag{28}$$

By applying Inequality equation 28 recursively, we derive the following bound

$$\mathrm{E}_{\xi_{[i-1,t-1]}}[f(w_{i-1,t}, \lambda_i) - f^*(\lambda_i)] \leq \left(1 - \frac{\mu}{L}\right)^k \mathrm{E}_{\xi_{[i-1]}}[f(w_{i-1}, \lambda_i) - f^*(\lambda_i)] \\ + \frac{\sigma^2}{2L}\sum_{j=0}^{k-1}\left(1 - \frac{\mu}{L}\right)^j. \tag{29}$$

Finally, by using the limit of geometric series, we obtain

$$\mathrm{E}_{\xi_{[i-1,t-1]}}[f(w_{i-1,t}, \lambda_i) - f^*(\lambda_i)] \leq \left(1 - \frac{\mu}{L}\right)^k \mathrm{E}_{\xi_{[i-1]}}[f(w_{i-1}, \lambda_i) - f^*(\lambda_i)] + \frac{\sigma^2}{2\mu}. \tag{30}$$

$\square$

## F    PROOF OF LEMMA 3.9

**Lemma F.1.** *Assume* $\|\lambda_{i+1} - \lambda_i\| \leq \epsilon$, $0 \leq \epsilon < \frac{B}{\delta+\gamma}$ *and let* $w_i$ *denote the i-th iterate of Algorithm 1 with* $\alpha \leq \frac{1}{L}$. *Under Assumptions 3.4- 3.3 and 3.5- 3.6, if* $\phi_{w_i}(\lambda_i) \leq B - (\delta+\gamma)\epsilon$, *then* $\phi_{w_i}(\lambda_{i+1}) \leq B$. *In addition, let*

$$k_{\max} := \left\lceil \log_\rho \left( 1 - \frac{2\mu(\delta+\gamma)\epsilon + \sigma^2}{2\mu B} \right) \right\rceil. \tag{31}$$

*If* $\phi_{w_i}(\lambda_{i+1}) \leq B$, $0 \leq \epsilon < \frac{1}{\delta+\gamma}\left(B - \frac{\sigma^2}{2\mu}\right)$ *and* $k \geq k_{\max}$, *then* $\phi_{w_{i+1}}(\lambda_{i+1}) \leq B - (\delta+\gamma)\epsilon$.

*Proof.* We start by deriving an upper bound on $\mathrm{E}_{\xi_{[i]}}[f(w_i, \lambda_{i+1})] - f^*(\lambda_{i+1})$ with $\phi_{w_i}(\lambda_i) \leq B$ and $\|\lambda_{i+1} - \lambda_i\| \leq \epsilon$. For that we use the regularity Assumptions 3.2 and 3.3 together with the triangle and Jensen inequalities as follows

$$
\begin{aligned}
\mathrm{E}_{\xi_{[i]}}[f(w_i, \lambda_{i+1})] - f^*(\lambda_{i+1}) &= |\mathrm{E}_{\xi_{[i]}}[f(w_i, \lambda_{i+1})] - f^*(\lambda_{i+1})| \\
&= \quad |\mathrm{E}_{\xi_{[i]}}[f(w_i, \lambda_{i+1}) + f(w_i, \lambda_i) - f(w_i, \lambda_i)] - f^*(\lambda_{i+1}) \\
&\qquad + f^*(\lambda_i) - f^*(\lambda_i)| \\
\overset{\text{Triangle and Jensen Ineq.}}{\leq} & \quad |\mathrm{E}_{\xi_{[i]}}[f(w_i, \lambda_i)] - f^*(\lambda_i)| + \mathrm{E}_{\xi_{[i]}}[|f(w_i, \lambda_{i+1}) - f(w_i, \lambda_i)|] \\
&\qquad + |f^*(\lambda_i) - f^*(\lambda_{i+1})| \\
\overset{\text{Assumptions 3.2 and 3.3}}{\leq} & \quad |\mathrm{E}_{\xi_{[i]}}[f(w_i, \lambda_i)] - f^*(\lambda_i)| + (\delta+\gamma)\epsilon.
\end{aligned}
\tag{32}
$$

From Inequality equation 32 it follows that, if $\phi_{w_i}(\lambda_i) \leq B - (\delta+\gamma)\epsilon$ with $\epsilon < \frac{B}{(\delta+\gamma)}$, then $\phi_{w_i}(\lambda_{i+1}) \leq B$.

We now use the results of Theorem 3.8 to derive a lower bound on the number of SGD-steps such that, if $\phi_{w_i}(\lambda_{i+1}) \leq B$, then $\phi_{w_{i+1}}(\lambda_{i+1}) \leq B - (\delta+\gamma)\epsilon$. We start considering the following inequality

$$
\begin{aligned}
\mathrm{E}_{\xi_{[i+1]}}[f(w_{i+1}, \lambda_{i+1})] - f^*(\lambda_{i+1}) &\leq \rho^k \left[ \mathrm{E}_{\xi_{[i]}}[f(w_i, \lambda_{i+1})] - f^*(\lambda_{i+1}) \right] + \frac{\sigma^2}{2\mu} \\
&\leq \rho^k B + \frac{\sigma^2}{2\mu}.
\end{aligned}
\tag{33}
$$

From Inequality equation 33 it follows that, if $\phi_{w_i}(\lambda_{i+1}) \leq B$, then $\phi_{w_{i+1}}(\lambda_{i+1}) \leq B - (\delta+\gamma)\epsilon$ with $\epsilon < \frac{1}{\delta+\gamma}\left(B - \frac{\sigma^2}{2\mu}\right)$ whenever

$$k \geq \left\lceil \log_\rho \left( 1 - \frac{2\mu(\delta+\gamma)\epsilon + \sigma^2}{2\mu B} \right) \right\rceil. \tag{34}$$

$\square$

## G    PROOF OF THEOREM 3.10

**Theorem G.1.** *Assume there exists* $\frac{\sigma^2}{2\mu} < r \leq B$ *and* $\tilde{\epsilon} := \min\{\epsilon_1, \epsilon_2\}$ *with*

$$\epsilon_1 := \frac{1}{(\delta+\gamma)}(B - r), \quad \epsilon_2 := \frac{(1 - \rho^k)r - \sigma^2/2\mu}{\rho^k(\delta+\gamma)}. \tag{35}$$

*In addition, let*

$$k_{\max} := \left\lceil \log_\rho \left( 1 - \frac{\sigma^2}{2\mu r} \right) \right\rceil. \tag{36}$$

*Consider Algorithm 1 with* $\alpha \leq \frac{1}{L}$, $k \geq k_{\max}$ *and* $\|\lambda_{i+1} - \lambda_i\| \leq \epsilon$, *where* $0 \leq \epsilon \leq \tilde{\epsilon}$.

*Under Assumptions 3.4- 3.3 and 3.5- 3.6, if* $\phi_{w_i}(\lambda_i) \leq r$, *then* $\phi_{w_{i+1}}(\lambda_{i+1}) \leq r$.

*Proof.* We consider $\phi_{w_i}(\lambda_i) \leq r$. If $\epsilon \leq \frac{1}{(\delta+\gamma)}(B-r)$ with $r \leq B$, then $\phi_{w_i}(\lambda_i) \leq B - (\delta+\gamma)\epsilon$ and, as shown in Lemma 3.9, $\phi_{w_i}(\lambda_{i+1}) \leq B$. This allows us to use the results of Theorem 3.8.

We now derive an upper bound on $\mathrm{E}_{\xi_{[i+1]}}[f(w_{i+1}, \lambda_{i+1})] - f^*(\lambda_{i+1})$ by considering the results of Theorem 3.8 together with the regularity Assumptions 3.2 and 3.3, and the triangle and Jensen inequalities as follows

$$
\begin{aligned}
\mathrm{E}_{\xi_{[i+1]}}[f(w_{i+1}, \lambda_{i+1})] - f^*(\lambda_{i+1}) &\leq \rho^k \left[ \mathrm{E}_{\xi_{[i]}}[f(w_i, \lambda_{i+1})] - f^*(\lambda_{i+1}) \right] + \frac{\sigma^2}{2\mu} \\
&= \rho^k \Big| \mathrm{E}_{\xi_{[i]}}[f(w_i, \lambda_{i+1}) + f(w_i, \lambda_i) - f(w_i, \lambda_i)] - f^*(\lambda_{i+1}) \\
&\qquad + f^*(\lambda_i) - f^*(\lambda_i) \Big| + \frac{\sigma^2}{2\mu} \\
\overset{\text{Triangle and Jensen Ineq.}}{\leq} &\quad \rho^k \left[ \mathrm{E}_{\xi_{[i]}}[f(w_i, \lambda_i)] - f^*(\lambda_i) \right] + \rho^k \mathrm{E}_{\xi_{[i]}}[|f(w_i, \lambda_{i+1}) - f(w_i, \lambda_i)|] \\
&\qquad + \rho^k |f^*(\lambda_i) - f^*(\lambda_{i+1})| + \frac{\sigma^2}{2\mu} \\
\overset{\text{Assumptions 3.2 and 3.3}}{\leq} &\quad \rho^k \left[ \mathrm{E}_{\xi_{[i]}}[f(w_i, \lambda_i)] - f^*(\lambda_i) \right] + \rho^k (\delta+\gamma)\|\lambda_{i+1} - \lambda_i\| + \frac{\sigma^2}{2\mu} .
\end{aligned}
\tag{37}
$$

Using the fact that $\phi_{w_i}(\lambda_i) \leq r$ and that $\|\lambda_{i+1} - \lambda_i\| \leq \epsilon$, we now solve the following inequality for $\epsilon$ in order to find an upper bound on the variation of the homotopy parameter such that $\phi_{w_{i+1}}(\lambda_{i+1}) \leq r$

$$
\rho^k\, r + \rho^k\,(\delta+\gamma)\,\epsilon + \frac{\sigma^2}{2\mu} \leq r .
\tag{38}
$$

Inequality equation 38 holds whenever

$$
k \geq \left\lceil \log_\rho \left( 1 - \frac{\sigma^2}{2\mu r} \right) \right\rceil ,
\tag{39}
$$

and

$$
\epsilon \leq \frac{(1-\rho^k)\, r - \sigma^2/2\mu}{\rho^k\,(\delta+\gamma)} ,
\tag{40}
$$

with $r > \frac{\sigma^2}{2\mu}$. $\qquad\square$

## H PROOF OF THEOREM 3.11

**Theorem H.1.** *Let $\tilde{\rho} \in \left( 1 - \frac{\sigma^2}{2\mu}\frac{1}{B}, 1 \right)$ and consider Algorithm 1 with $\alpha \leq \frac{1}{L}$, $\phi_{w_0}(\lambda_0) \leq r$ with $\frac{\sigma^2}{2\mu}\frac{1}{(1-\tilde{\rho})} \leq r \leq B$ and $k \geq \log_\rho(\tilde{\rho})$. In addition, let $\epsilon_1 := \frac{1}{(\delta+\gamma)}(B-r)$ and*

$$
C_{\tilde{\rho}} := \begin{cases} 1 & \text{if } k \geq \log_\rho(\tilde{\rho}) - \log_\rho\left(1 + \frac{\delta+\gamma}{\varepsilon_0}\right) \\ \frac{\tilde{\rho}-\rho^k}{\rho^k} \frac{\varepsilon_0}{(\delta+\gamma)} & \text{otherwise,} \end{cases}
\tag{41}
$$

*with $\varepsilon_0 := \mathrm{E}_{\xi_{[0]}}[f(w_0, \lambda_0)] - f^*(\lambda_0)$.*

*Under Assumptions 3.4- 3.3 and 3.5- 3.6, if $\|\lambda_{i+1} - \lambda_i\| \leq \min\left\{e^{-\eta i}, \epsilon_1\right\}$ with $\eta \geq \ln\left(C_{\tilde{\rho}}\,\tilde{\rho}\right)$, then*

$$
\mathrm{E}_{\xi_{[i+1]}}[f(w_{i+1}, \lambda_{i+1})] - f^*(\lambda_{i+1}) \leq \tilde{\rho}^{i+1}\left[ \mathrm{E}_{\xi_{[0]}}[f(w_0, \lambda_0)] - f^*(\lambda_0) \right] + \frac{\sigma^2}{2\mu}\sum_{j=0}^{i} \tilde{\rho}^j .
\tag{42}
$$

*Proof.* We start assuming that $\phi_{w_i}(\lambda_i) \leq r$, with $0 \leq r \leq B$ and $\|\lambda_{i+1} - \lambda_i\| \leq \epsilon_1$ with $\epsilon_1 := \frac{1}{(\delta+\gamma)}(B-r)$ such that $\phi_{w_i}(\lambda_{i+1}) \leq B$.

In particular, we consider the following upper bound on $\mathrm{E}_{\xi_{[i+1]}}\left[f(w_{i+1}, \lambda_{i+1})\right] - f^*(\lambda_{i+1})$,

$$\mathrm{E}_{\xi_{[i+1]}}\left[f(w_{i+1}, \lambda_{i+1})\right] - f^*(\lambda_{i+1}) \leq \rho^k \left[\mathrm{E}_{\xi_{[i]}}\left[f(w_i, \lambda_i)\right] - f^*(\lambda_i)\right]$$
$$+ \rho^k \left(\delta + \gamma\right) \Delta\lambda_{i+1} + \frac{\sigma^2}{2\mu}. \tag{43}$$

See the the proof of Theorem 3.10 for a derivation.

We then proceed by induction. Therefore, we assume

$$\mathrm{E}_{\xi_{[i]}}\left[f(w_i, \lambda_i)\right] - f^*(\lambda_i) \leq \tilde{\rho}^i \left[\mathrm{E}_{\xi_{[0]}}\left[f(w_0, \lambda_0)\right] - f^*(\lambda_0)\right] + \frac{\sigma^2}{2\mu} \sum_{j=0}^{i-1} \tilde{\rho}^j, \tag{44}$$

and derive the conditions on $\Delta\lambda_{i+1}$ such that

$$\mathrm{E}_{\xi_{[i+1]}}\left[f(w_{i+1}, \lambda_{i+1})\right] - f^*(\lambda_{i+1}) \leq \tilde{\rho}^{i+1} \left[\mathrm{E}_{\xi_{[0]}}\left[f(w_0, \lambda_0)\right] - f^*(\lambda_0)\right] + \frac{\sigma^2}{2\mu} \sum_{j=0}^{i} \tilde{\rho}^j. \tag{45}$$

In order to achieve that, we consider the upper bound on $\mathrm{E}_{\xi_{[i+1]}}\left[f(w_{i+1}, \lambda_{i+1})\right] - f^*(\lambda_{i+1})$ given by Inequality equation 43 and solve the following inequality for $\Delta\lambda_{i+1}$

$$\rho^k \left[\mathrm{E}_{\xi_{[i]}}\left[f(w_i, \lambda_i)\right] - f^*(\lambda_i)\right] + \rho^k \left(\delta + \gamma\right) \Delta\lambda_{i+1} + \frac{\sigma^2}{2\mu}$$
$$\leq \tilde{\rho}^{i+1} \left[\mathrm{E}_{\xi_{[0]}}\left[f(w_0, \lambda_0)\right] - f^*(\lambda_0)\right] + \frac{\sigma^2}{2\mu} \sum_{j=0}^{i} \tilde{\rho}^j. \tag{46}$$

We obtain that Inequality equation 46 is satisfied whenever

$$\Delta\lambda_{i+1} \leq \underbrace{\frac{\tilde{\rho}^{i+1} \left[\mathrm{E}_{\xi_{[0]}}\left[f(w_0, \lambda_0)\right] - f^*(\lambda_0)\right] - \rho^k \left[\mathrm{E}_{\xi_{[i]}}\left[f(w_i, \lambda_i)\right] - f^*(\lambda_i)\right] + \frac{\sigma^2}{2\mu} \sum_{j=1}^{i} \tilde{\rho}^j}{\rho^k(\delta + \gamma)}}_{\mathrm{RHS}}. \tag{47}$$

We derive a lower bound on the right-had side of equation 47 by considering the induction assumption, i.e. Inequality equation 44, and we obtain

$$\mathrm{RHS} \geq \frac{\tilde{\rho}^i \left(\tilde{\rho} - \rho^k\right) \left[\mathrm{E}_{\xi_{[0]}}\left[f(w_0, \lambda_0)\right] - f^*(\lambda_0)\right] - \frac{\sigma^2}{2\mu}\rho^k \sum_{j=0}^{i-1} \tilde{\rho}^j + \frac{\sigma^2}{2\mu} \sum_{j=1}^{i} \tilde{\rho}^j}{\rho^k(\delta + \gamma)}. \tag{48}$$

Considering that $k \geq \log_\rho(\tilde{\rho})$ then

$$-\frac{\sigma^2}{2\mu}\rho^k \sum_{j=0}^{i-1} \tilde{\rho}^j + \frac{\sigma^2}{2\mu} \sum_{j=1}^{i} \tilde{\rho}^j = \frac{\sigma^2}{2\mu} \left(1 - \frac{\rho^k}{\tilde{\rho}}\right) \sum_{j=0}^{i-1} \tilde{\rho}^j \geq 0. \tag{49}$$

Consequently, Inequality equation 47 is satisfied whenever

$$\Delta\lambda_{i+1} \leq \frac{\left(\tilde{\rho} - \rho^k\right)}{\rho^k} \frac{\varepsilon_0}{(\delta + \gamma)} \tilde{\rho}^i, \tag{50}$$

where $\varepsilon_0 \coloneqq \mathrm{E}_{\xi_{[0]}}\left[f(w_0, \lambda_0)\right] - f^*(\lambda_0)$.

To conclude this first part of derivations, we obtain that, whenever Inequality equation 50 is satisfied, then

$$\mathrm{E}_{\xi_{[i+1]}}\left[f(w_{i+1}, \lambda_{i+1})\right] - f^*(\lambda_{i+1}) \leq \tilde{\rho}^{i+1} \left(\mathrm{E}_{\xi_{[0]}}\left[f(w_0, \lambda_0)\right] - f^*(\lambda_0)\right) + \frac{\sigma^2}{2\mu} \sum_{j=0}^{i} \tilde{\rho}^j. \tag{51}$$

In order to ensure that $\phi_{w_{i+1}}(\lambda_{i+1}) \leq r$ we consider Inequality equation 51 and use the fact that $\phi_{w_0}(\lambda_0) \leq r$, i.e. $\mathrm{E}_{\xi_{[0]}}\left[f(w_0, \lambda_0)\right] - f^*(\lambda_0) \leq r$, to upper bound the right-hand side of Inequality equation 51. We then solve the resulting inequality for $r$

$$\tilde{\rho}^{i+1} r + \frac{\sigma^2}{2\mu} \sum_{j=0}^{i} \tilde{\rho}^j \leq r. \tag{52}$$

Considering that $\sum_{j=0}^{i} \tilde{\rho}^{j} = \frac{\left(1 - \tilde{\rho}^{i+1}\right)}{(1 - \tilde{\rho})}$, we obtain that Inequality equation 52 is satisfied whenever

$$r \geq \frac{\sigma^2}{2\mu} \frac{1}{(1 - \tilde{\rho})} \, . \tag{53}$$

By combining Inequality equation 53 with the fact that $r \leq B$, we obtain the following upper bound on $\tilde{\rho}$

$$\tilde{\rho} \leq 1 - \frac{\sigma^2}{2\mu} \frac{1}{B} \, . \tag{54}$$

To further simplify the bound in equation 50, we define the following constant

$$C_{\tilde{\rho}} := \begin{cases} 1 & \text{if } k \geq \log_\rho(\tilde{\rho}) - \log_\rho\left(1 + \frac{\delta + \gamma}{\varepsilon_0}\right) \\ \frac{\tilde{\rho} - \rho^k}{\rho^k} \frac{\varepsilon_0}{(\delta + \gamma)} & \text{otherwise,} \end{cases} \tag{55}$$

and obtain that Inequality equation 50 holds whenever $\Delta \lambda_{i+1} \leq e^{-\eta\,i}$ with $\eta \geq -\ln\left(C_{\tilde{\rho}}\,\tilde{\rho}\right)$. $\qquad\square$

