# OpenReview forum: "Convergence Analysis of Homotopy-SGD for Non-Convex Optimization"
_ICLR.cc/2021/Conference — Reject_

### Official Review · AnonReviewer1 · 2020-10-28
**Interesting idea, but motivation, presentation and comparison with prior work need major improvements.**

**Rating:** 5
**Confidence:** 4

**Review:**

This paper proposes homotopy SGD (H-SGD) which solves a sequence of unconstrained problems with a homotopy map and homotopy parameter. The authors analyze the algorithm for solving nonconvex problems satisfying PL condition. The analysis works with a generic homotopy map and homotopy parameter satisfying certain conditions (given in Sec 3.1). The authors show linear convergence to a neighborhood of the minimizer. The theoretical results are validated with experiments with clear explanations.

In Gargiani et al, 2020, it was shown that using homotopy with SGD is a useful approach for transfer learning. This paper paper continues along this line, and relaxes the assumption of local strong convexity used in Gargiani et al, 2020 to PL inequality.

This being said, I found the writing of the paper unclear, making the motivation of the paper also unclear. For example, the paper keeps comparing H-SGD with SGD throughout. But in the regular optimization setting, it is not clear why one needs to utilize homotopy. For example, in page 3 it is written "given our setting, vanilla SGD can only ensure a global sublinear rate of convergence". First, what is the reference for this claim? Second, I think with the PL assumption, it is well known that SGD gets linear convergence to a neighborhood of the minimizer (In Vaswani et al, 2019, proof of Thm 4, we will not let $\sigma=0$, but keep it, which will determine the neighborhood to which the algorithm converges linearly).

On the other hand, in Sec 4.1 the authors explain that even if PL holds globally, the constant might be getting worse as moving to the solution, so SGD suffers from the worst case constant in the rate and H-SGD has a better "conditioning". I think such explanations are useful and should be given throughout.

In terms of the assumption, the authors claim in Remark C.2 that it is weaker than PL of Vaswanit et al, 2019. In the proof of Prop 3.7, the authors use law of iterated expectation to iterate the expectation one step and use the expected PL condition. Why can't the same be done in the proof of Thm 4 of Vaswani et al, 2019? From what I see, the same assumption can be used in Vaswani et al, 2019 to get the same result with SGD. Can the authors clarify this part?

Another point about Assumption 3.6 is the following: the authors state in page 5 that compared to PL, this assumption is local. If the assumption is local, this means that convergence results of the paper will also be local. The natural question will then be, what does H-SGD do until it reaches to local region where the expected PL inequality holds?

Assumption 3.1 is quite unclear, and the notations are undefined. Given that the assumption is not explained in words, it makes it quite difficult to understand what it means. Moreover, what is $z$ that is used in Assumption 3.1? I did not see it defined anywhere. Could the authors be more explicit here, by defining the notation used in Assumption 3.1 (for example $U^*(\lambda)$ and $z$).

In Alg.1, $h(i)$ needs to be chosen such that its sum from i=1 to n will be 1. Does it mean that one needs to know $n$ to run the algorithm? If so, then this will require setting an horizon before running the algorithm. Is there a way to ensure this "sum to 1" condition holds, adaptively?

Assumptions given in Sec 3.1 are quite vague, is it possible to give examples? For example, the authors might add some examples of homotopy map and show if the assumptions are satisfied.

In the experimental part, only toy examples are considered. I would suggest the authors to include more "real-life" problems to show the merit of the new approach in practice.

Minor comment: Is the algorithm of https://arxiv.org/abs/1902.00126 a special case of H-SGD? The idea in that paper seems similar.

Overall, I think that presenting H-SGD as an alternative to SGD makes the motivation of the method unclear. Moreover, the notation and explanations for the assumptions, theorems are missing and necessary for judging the significance of the results, compared to related work. As a follow-up on Gargiani et al, 2020, I find the paper interesting, however, the assumptions given in Sec 3.1 should be compared with Gargiani et al, 2020 clearly. In the current situation, the presentation of the paper is problematic, which in my opinion, also shadows the concrete contribution. If the authors clarify my questions, I can reconsider my score.

============ after discussion phase =============

My important questions about locality of PL are not explained. For example, on why the same locality argument cannot be done on Vaswani et al, 2019's analysis on standard SGD. As I also stressed in my original review, I believe the idea of the paper is interesting and can be useful, however the merit of the paper is not explained clearly in the paper. Rather than comparing by SGD with vague arguments, I think the authors should clearly explain under what setting is homotopy preferable to SGD and why, which will make the paper much more accessible and impactful. Given the lack of explanations, unfortunately, I keep my score for rejection.

---

> ### Author Response · Authors · 2020-11-17
> **Thank you for your review - we think that some key aspects have been misunderstood.**
>
> We thank the reviewer for taking the time to read our submission and provide feedback. However, we fear that some very fundamental aspects of the problem statement have been misunderstood.
>
> Regarding your doubts on the paper's motivation:
>
> "Therefore, the ideal scenario would be to be able to exploit the stronger local structure while the method’s iterates gradually approach a minimizer and independently from the starting point. In this regard, homotopy methods are a general strategy for tackling difficult optimization problems by gradually transforming a simplified version of a target problem, or a version with a known minimizer, back to its original form while following a solution along the way. Consequently, they preserve in each step the vicinity to a minimizer of the currently tackled problem, allowing the solver to always work in regions where the problems exhibit
> stronger structures."
>
> Regarding your comment on the PL condition, namely:
>
> For example, in page 3 it is written "given our setting, vanilla SGD can only ensure a global sublinear rate of convergence". First, what is the reference for this claim? Second, I think with the PL assumption, it is well known that SGD gets linear convergence to a neighborhood of the minimizer (In Vaswani et al, 2019, proof of Thm 4, we will not let $\sigma=0$, but keep it, which will determine the neighborhood to which the algorithm converges linearly)
>
> we would like to stress that we only assume that the PL condition holds **locally** and therefore SGD can only enjoy linear convergence when the iterates are starting in such local region and not from any starting point. As we show with our theoretical analysis, H-SGD enjoys linear convergence in such scenario even when starting far from a minimizer of the target problem.

---

### Official Review · AnonReviewer2 · 2020-10-28
**This algorithm is not that practical and the the hypothesis is a bit vague.**

**Rating:** 4
**Confidence:** 4

**Review:**

1. It seems to me the proposed Homotopy-SGD is not a practical algorithm, as in each iteration the algorithm has to solve a nontrivial (possibly nonconvex) subproblem. In other words, each subproblem can be as difficult as the original problem. This leads to an essential question that what is the practical motivation of this algorithm?

2. Algorithm 1 is not complete. The authors should explicitly write down Step 6: w_i \leftarrow SGD(.)

3. I do not understand the notation [0,1]^z.

4. I do not understand why do you define W*(\lambda) in Assumption 3.1, it seems to be exactly the same as U*(\lambda). It looks this assumption is nothing else than assuming the set of minimizers of each subproblem is nonempty. If it is, I would suggest making the statement simpler.

5. What is the difference between Assumption 3.2 and 3.3?

6. I do not agree that Assumption 3.6 (the key assumption for local convergence rate analysis) is weaker than the original PL condition. I agree that the global PL condition is quite strong for a nonconvex problem and is indeed unrealistic. But this is not the reason that your assumption is weaker. Note that you involve expectation over the past random samplings and algorithmic iterates in this assumption. How can one guarantee that the algorithmic trajectory should satisfy a specific function growth condition? A function regularity should always be stated with respect to the problem itself and should be independent of the algorithm. That is, it should be a problem-intrinsic property. Typical examples include Lipschitz continuity, quadratic growth, strong convexity, etc. On the other hand, I tend to think the expectation used in this assumption is also tailored for the analysis. If the authors assume a local version of the PL condition, the first difficulty would be to show the iterates of H-SGD method stay within this local region (in which the PL condition holds). It is because the convergence result is in expectation (due to the randomness of the algorithm), in order to use a deterministic local PL condition, a natural way is to bound the iterates within this local region with high probability, but this is often the hardest part.

7. The convergence result seems to be a local one as you have to assume a good initialization, which you should explicitly state in abstract and introduction rather than saying ‘H-SGD can achieve a global linear rate of convergence’. Also, this good initialization requirement is related to my last comments on your assumption 3.6. You directly assume the algorithm can be initialized within a local region in expectation and the then ‘expected PL condition’ can help H-SGD to make a good progress towards the minimizer. Lastly, local convergence can be established. However, as I commented above, these assumptions are too tailored and stringent.

8. I do not see why Theorem 3.11 implies a linear rate of convergence. The last term in (14) is divergent when i tends to infinity.

9. I suggest change ‘problem 1’ to ‘problem (1)’ globally.

---

> ### Author Response · Authors · 2020-11-17
> **Thank you for your review - we reply to some of your points**
>
> Thank you for your review. We think that some aspects of our work have been misunderstood.
> We answer to some of your points:
>
> 1. It seems to me the proposed Homotopy-SGD is not a practical algorithm, as in each iteration the algorithm has to solve a nontrivial (possibly nonconvex) subproblem. In other words, each subproblem can be as difficult as the original problem. This leads to an essential question that what is the practical motivation of this algorithm?
>
> We specify multiple times and in the introduction as well that the problems are solved **approximately**, i.e., with a limited number of SGD iterations on each instance.
> "By using such a homotopy map, H-SGD finds an approximate solution of Problem 1 by **approximately** solving a series of parametric problems that gradually leads to the target one."
>
> 3. I do not understand the notation [0,1]^z
>
> it is a standard notation for the n-ary cartesian product. Please see https://en.wikipedia.org/wiki/Cartesian_product. E.g. [0,1]^2 = [0,1]\times [0,1].
>
> 5. What is the difference between Assumption 3.2 and 3.3?
>
> We are not sure what the reviewer means. Assumptions 3.2 and 3.3 are structurally different assumptions, see their definition. Assumption 3.2 regards the objective functions while 3.3 the optimal value function.
>
> 6. I do not agree that Assumption 3.6 (the key assumption for local convergence rate analysis) is weaker than the original PL condition. I agree that the global PL condition is quite strong for a nonconvex problem and is indeed unrealistic. But this is not the reason that your assumption is weaker. Note that you involve expectation over the past random samplings and algorithmic iterates in this assumption. How can one guarantee that the algorithmic trajectory should satisfy a specific function growth condition? A function regularity should always be stated with respect to the problem itself and should be independent of the algorithm. That is, it should be a problem-intrinsic property. Typical examples include Lipschitz continuity, quadratic growth, strong convexity, etc. On the other hand, I tend to think the expectation used in this assumption is also tailored for the analysis. If the authors assume a local version of the PL condition, the first difficulty would be to show the iterates of H-SGD method stay within this local region (in which the PL condition holds). It is because the convergence result is in expectation (due to the randomness of the algorithm), in order to use a deterministic local PL condition, a natural way is to bound the iterates within this local region with high probability, but this is often the hardest part.
>
> Please see https://arxiv.org/pdf/2006.10311.pdf and https://arxiv.org/pdf/1805.02632.pdf for analogous assumptions.
>
> 8. I do not see why Theorem 3.11 implies a linear rate of convergence. The last term in (14) is divergent when i tends to infinity.
> That is a geometric series with argument less than one in absolute value, therefore it does not tend to infinity. Please see https://en.wikipedia.org/wiki/Geometric_series.

---

### Official Review · AnonReviewer3 · 2020-10-29
**Summary**

**Rating:** 5
**Confidence:** 4

**Review:**

This paper proposed a Homotopy-Stochastic Gradient Descent (H-SGD) algorithm by applying homotopy strategy to explore the nice local structures of problems. H-SGD can gradually approximate to the target objective function and enjoys a global linear convergence to reach a neighborhood of a minimizer. As verified by the author, the assumption of this paper is weaker than its predecessors, Karimi et al., 2016; Vaswani et al., 2019. Further, the numerical experiments verified the effectiveness of H-SGD on regression and classification tasks.
However, there are still some concerns about this paper:
Recently, people are focusing on investigating convergence to achieve the global minimizer or $\epsilon$-stationary points. It is quite novel that authors brought the convergence to a sub-level set up. However, by the main theorem 3.11 of the paper, the sublevel set is $O(\frac{1}{u})$. It still remains unclear whether it is acceptable as $\mu$ could be $1e-6$, $1e-7$ as shown by the author. Further, achieving the global linear convergence to a sublevel set is not new, it is even achievable for the widely used SGD with momentum for the quadratic function without any assumptions on it.
On the other hand, the author didn’t provide a detailed formulation (examples) of $f(w,\lambda)$ throughout the whole paper, which would increase the difficulty for the reader to understand. Suppose the loss $f_i(\w)$ satisfies the assumption of $\w$, then losses with the l2_morn (l2_norm^2) regularization,  $f(\w,\lambda) = 1/N\sum\limits_{i=1}^N f_i(\w)  + \frac{\lambda}{2}\|\w\|^2$ satisfies the assumptions of the paper as long as $||\w||$ is bounded. The global convergence to the optimal solution which also has been well studied in other literatures with or without PL condition, such as,https://arxiv.org/pdf/1812.03934.pdf. As reaching a neighborhood of a minimizer is an unavoidable step to reach the global minimizer, the theory contribution of the paper is required to be considered more carefully.
Lastly, the numerical experiments are quite limited. https://arxiv.org/pdf/1811.03962.pdf  has shown that deep neural networks with relu activation, such as resnets, satisfies PL condition. The effectiveness of H-SGD would be more persuasive if the experiments could be conducted on real data sets (cifar, imagenets, etc) using well recognized backbones (resnets, etc) even though it probably violates the assumption to some degree..

---

> ### Author Response · Authors · 2020-11-17
> **Thank you for your review - the authors are struggling to understand your comments on global optimization methods**
>
> We thank you for your review. Unfortunately we struggle to understand the relationship between global optimization methods and our work, whose goal is to efficiently compute a local minimizer.

---

### Official Review · AnonReviewer5 · 2020-11-06
**The idea is nice, however, the theory is incomplete and the baselines are not outperformed**

**Rating:** 5
**Confidence:** 4

**Review:**


Summary:

The paper proposes a homotopy approach for minimizing finite sum optimization problems. The idea behind HSGD is to gradually solve a more and more similar problem to the original one using SGD, and always use the approximate solution of the previous problem as a good starting point to solve the new one.



Main reason to accept the paper :
I believe that homotopy is a nice idea that is still under-explored in the optimization/machine learning literature. I find the topic to be interesting for the ICLR community.
To the best of my knowledge, H-SGD is a novel (in the form that it is stated).


Main reason to reject the paper:

The main problem I have with this paper is that HSGD is not justified to outperform vanilla SGD in any scenario.
Besides that, nothing else about this paper stands out, and a lot of other criticism can be made (see detailed comments).


Overall, I believe that the paper's contributions are not substantial enough for acceptance, and thus I'm recommending to reject this paper. I am willing to change my evaluation, given that the authors persuade me that the provided theory of HSGD is indeed better than the theory of vanilla SGD. (see points 3b, 5,6 below)



Detailed comments:

1) Proof of Proposition 3.7 seems incorrect/incomplete. I have a problem with the last paragraph of the proof, which does not justify the proposition at all. I have checked myself and the proposition indeed holds as a consequence of (24); please fix the proof.

2) Theorem 3.8.: This is nothing new, but rather a standard analysis of SGD under the uniformly bounded variance. Please reference.

3) There are a few misleading claims throughout the paper.
3a) First of all, the paper sells the application as a nonconvex optimization while only providing the results in the PL setting. Note that PL is rather a generalization of the strong convexity; the class of PL problems is significantly rather similar to the strongly convex problems than the general nonconvex objectives. In fact, the whole motivation of PL inequality was to find the broader class of problems where one can get a strongly convex-like rate. I thus find misleading to present the paper as a nonconvex optimization.
3b) It is claimed that Homotopy SGD converges linearly to the neighborhood of the optimum, while "vanilla SGD can only ensure a global sublinear rate of convergence". A similar claim is made in the abstract too. This is simply not true. Vanilla SGD converges linearly to a neighborhood of the optimum as well! This is even proven in the paper; Theorem 3.8 provides such a rate with $\lambda=1$.

4) Assumptions are very strong. Specifically, the boundedness of the variance is very rarely satisfied in practice and is not required for the state-of-the-art SGD analysis under relaxed, strong convexity [1,2]. Note that in certain scenarios, one should not assume bounded variance and strong convexity (or its realizations) at the same time as it significantly shrinks the class of functions [1].

5) The theory is not complete. It would be great to have to state what exactly the complexity of HSGD is, namely, how many stochastic gradients in total one needs to get to some
specific neighborhood of the optimum of (1). Without such a result, one can not argue that HSGD is better than any baseline, such as vanilla SGD. In fact, I do believe that the overall rate of HSGD would be inferior to the rate of vanilla SGD. Note that Thm 3.11 does not provide anything close to it, as it gives a suboptimality of $f(x,\lambda_i)$ only (we want $\lambda =1$).

6) Following the point 3b) and 5), I do not see any advantage (in theory) of homotopy SGD over classical vanilla SGD. There is maybe only one -- while vanilla SGD and HSGD require a PL condition among a different set of points; there is thus a chance that the "empirical PL" would be better for the HSGD. However, one can not know this in advance; and one can not know this even during the run of the algorithm. Further, the current theory does not allow to exploit "better" PL constant in certain areas of the objective since Assumption 3.6 considers a single $\mu$ throughout $R^d\times [0,1]$ (vanilla
SGD requires PL to hold only over $R^d$ so it is even less restrictive).

7) An approach similar to homotopy optimization (gradually solving easier problem instances, which could be presented as homotopy optimization if one wanted to) already made a significant impact in the optimization field. Specifically, it was shown that such an approach -- Catalyst -- might accelerate (in the sense of Nesterov) almost any optimization algorithm [3].

8) Please add some comments next to the assumptions explaining how strong each of those is.

9) Correctness: I did a detailed check of some results and a high-level check of the rest. I did not find any major or non-fixable flaws; the obtained results are reasonable.

10) Experiments: Experiments are not very strong either. HSGD is shown to outperform SGD in three toy examples only (often no by a large enough margin); this is not enough.

11) Question: Why do you limit yourself to the homotopy SGD method? I see no reason why the homotopy method can not be coupled with other optimizers; i.e., one can do SGD without bounded gradients, variance reductions, acceleration, or possibly even second-order methods. I also see that some of these methods would directly fit into the proposed homotopy theory.



[1] Nguyen, Lam M., et al. "SGD and Hogwild! convergence without the bounded gradients assumption." arXiv preprint arXiv:1802.03801 (2018).

[2] Gower, Robert Mansel, et al. "SGD: General analysis and improved rates." ICML 2019.

[3] Lin, Hongzhou, Julien Mairal, and Zaid Harchaoui. "A universal catalyst for first-order optimization." Advances in neural information processing systems. 2015.


************EDIT***************
I have raised my score to "5" after the author's response. While now I believe that once can come up with a scenario where the proposed theory of Homotopy SGD outperforms the vanilla SGD, it is still not properly demonstrated in the paper; there are a lots of hidden strings attached to the provided convergence bound (explained in my response).

---

> ### Author Response · Authors · 2020-11-17
> **Thank you for your insightful comments - we reply to points 3a, 3b, 5, 6**
>
> Thank you for your insightful comments - we reply to points 3a, 3b, 5, 6 in the hope that you will reconsider our work.
>
> 3a) First of all, the paper sells the application as a nonconvex optimization while only providing the results in the PL setting. Note that PL is rather a generalization of the strong convexity; the class of PL problems is significantly rather similar to the strongly convex problems than the general nonconvex objectives. In fact, the whole motivation of PL inequality was to find the broader class of problems where one can get a strongly convex-like rate. I thus find misleading to present the paper as a nonconvex optimization.
>
> PL condition is assumed to hold only **locally** and therefore it is a realistic assumption also for non-convex landscapes. The point of using a homotopy method is indeed to exploit stronger **local** structures by always operating in neighborhood of minimizers across the different homotopy problems.
>
> 3b) It is claimed that Homotopy SGD converges linearly to the neighborhood of the optimum, while "vanilla SGD can only ensure a global sublinear rate of convergence". A similar claim is made in the abstract too. This is simply not true. Vanilla SGD converges linearly to a neighborhood of the optimum as well! This is even proven in the paper; Theorem 3.8 provides such a rate with $\lambda=1$.
>
> vanilla SGD attains a **global** sublinear rate of convergence under the considered assumptions,  namely when the iterates start outside the local PL region. Therefore vanilla SGD converges linearly to a neighborhood of the optimum as well but only when the iterations are started in the local PL region for the target problem.
>
> 5. The theory is not complete. It would be great to have to state what exactly the complexity of HSGD is, namely, how many stochastic gradients in total one needs to get to some specific neighborhood of the optimum of (1). Without such a result, one can not argue that HSGD is better than any baseline, such as vanilla SGD. In fact, I do believe that the overall rate of HSGD would be inferior to the rate of vanilla SGD. Note that Thm 3.11 does not provide anything close to it, as it gives a suboptimality of $f(x,\lambda_i)$ only (we want $\lambda =1$).
>
> The goal of the theoretical analysis is not that of showing that H-SGD is always better than vanilla SGD, but just that can attain a linear rate of convergence even when the iterates are starting outside the local PL region on the target problem. Given our reply to your comment 3b, since vanilla SGD has sublinear convergence rate. For empirical evidence of the advantages of H-SGD, please see the Figures in our appendix.
>
> 6. Following the point 3b) and 5), I do not see any advantage (in theory) of homotopy SGD over classical vanilla SGD. There is maybe only one -- while vanilla SGD and HSGD require a PL condition among a different set of points; there is thus a chance that the "empirical PL" would be better for the HSGD. However, one can not know this in advance; and one can not know this even during the run of the algorithm. Further, the current theory does not allow to exploit "better" PL constant in certain areas of the objective since Assumption 3.6 considers a single $\mu$ throughout $R^d\times [0,1] (vanilla SGD requires PL to hold only over $R^d$ so it is even less restrictive).
>
> We only require the PL condition to hold locally, while vanilla SGD requires it to hold globally in order to achieve a linear rate of convergence when starting with an iterate that is arbitrarily far from a minimizer.

---

> > ### Comment · AnonReviewer5 · 2020-11-25
> > **The locality of PL**
> >
> > I would like to thank the authors for the response. Some points from my review were addressed and it was argued that the main contribution of the paper is valid. I will re-evaluate my score.
> >
> > I admit that the role of the locality of PL condition plays a more important role than what I believed; and at the same time, this is stressed well enough in the paper. I agree with the authors that Homotopy SGD might potentially outperform vanilla SGD given that PL condition holds locally only.
> >
> > However, some of the criticism is still valid. First of all, the theory is still not complete. Namely, one is interested to know how many stochastic gradients in total one needs to get to some specific neighborhood of the optimum of (1), which is not provided. If such a result was provided, you could easily argue that Homotopy SGD is indeed better than any other reasonable baseline.
> >
> > Next, even if $f(x,1)$ is locally PL with constant $\mu$ on a desired neighborhood, the construction from the paper does not guarantee that $f(x,\lambda)$ is PL too on a desired neighborhood of the optimum. It would be great to see an example where this actually holds so that indeed it is clear that the provided theory of Homotopy SGD can indeed outperform vanilla SGD in theory.
> >
> > Going further, the region where PL condition holds is not an "user-defined" constant; is assumed to be at least $
> > \frac{\sigma^2}{\mu}$ which is rather large. Such a requirement would not be needed if a different algorithm was considered instead of SGD.
> >
> > I strongly suggest that the authors give a simple and concrete example where Homotopy SGD outperforms Vanilla SGD according to theory and provided convergence bounds. While I believe that this should be possible (in contrast to when I was writing the initial review), it is not shown in the current version of the paper.

---

> > > ### Author Response · Authors · 2020-11-25
> > > **Thank you for your insightful comments and for having revisited your grade based on our reply.**
> > >
> > > We would like to thank the reviewer once again for the insightful comments. We are going to consider her/his suggestions for an extension of this work.

---

### Decision · Program_Chairs · 2021-01-07
**Final Decision**

**Decision:**

Reject

**Comment:**

The authors provide a homotopy framework for SGD in order to exploit structures that arise by construction, such as PL. I very much liked the delineated homotopy analysis which is general (i.e., as opposed to simply adding a quadratic, the authors consider a homotopy mapping). While the algorithm should not be considered new, it is still a good proposal to consider in the SGD applications setting. Unfortunately, I cannot recommend acceptance because of several issues that the reviewers raised in detail: Strength of the assumptions, unclear performance improvement in practice, applicability of the locally PL condition, among others.